# Novel determinants of NOTCH1 trafficking and signaling in breast epithelial cells

Francis M Kobia[1] , Luis Castro e Almeida[1] , Alyssa JJ Paganoni[1] , Francesca Carminati[1], Adrian Andronache[2], Francesco Lavezzari[1] , Mark Wade[2], Thomas Vaccari[1]

**The evolutionarily conserved Notch signaling pathway controls cell–cell communication, enacting cell fate decisions during development and tissue homeostasis. Its dysregulation is associated with a wide range of diseases, including congenital disorders and cancers. Signaling outputs depend on maturation of Notch receptors and trafficking to the plasma membrane, endocytic uptake and sorting, lysosomal and proteasomal degradation, and ligand-dependent and independent proteolytic cleavages. We devised assays to follow quantitatively the trafficking and signaling of endogenous human NOTCH1 receptor in breast epithelial cells in culture. Based on such analyses, we executed a high-content screen of 2,749 human genes to identify new regulators of Notch that might be amenable to pharmacologic intervention. We uncovered 39 new NOTCH1 modulators for NOTCH1 trafficking and signaling. Among them, we find that *PTPN23* and *HCN2* act as positive NOTCH1 regulators by promoting endocytic trafficking and NOTCH1 maturation in the Golgi apparatus, respectively, whereas *SGK3* serves as a negative regulator that can be modulated by pharmacologic inhibition. Our findings might be relevant in the search of new strategies to counteract pathologic Notch signaling.**

## Introduction

The evolutionarily conserved Notch pathway is a form of direct cell–cell communication that is extensively deployed in the regulation of multiple cellular functions during development and in adults. It influences cell fate decisions, cell proliferation and cell differentiation, and contributes to the maintenance of normal tissue homeostasis in both invertebrate and vertebrate metazoans (Artavanis-Tsakonas et al, 1999; Andersson et al, 2011; Koch et al, 2013). Because of such broad function, loss or gain of Notch function is associated with several congenital disorders (Mašek & Andersson, 2017) and cancers (Aster et al, 2017).

Upon synthesis in the ER, Notch proteins are processed at site S1 by Furin in the TGN (trans-Golgi network), the distal portion of Golgi apparatus (GA). Such cleavage generates the mature receptors exposed on the cell surface that are held together by $Ca^{2+}$ coordination within the heterodimerization domain (HD; see Fig 1A for a schematic of human NOTCH1) (Blaumueller et al, 1997; Rand et al, 2000). Once at the cell surface, Notch receptor activation can be initiated by binding the Delta/Serrate/Lag2 (DSL) family of Notch ligands that are present on the surfaces of adjacent signal-sending cells (Kopan & Ilagan, 2009). Following ligand binding, the receptors undergo two activating proteolytic cleavages; the first one is extracellular and is executed by ADAM10 at site S2 (van Tetering et al, 2009). The second is intracellular and is mediated at site S3 by the γ-secretase complex (Herreman et al, 2000; Schroeter et al, 1998; Zhang et al, 2000) (see Fig 1A for a schematic of human NOTCH1). These cleavages generate the transcriptionally competent Notch Intracellular Domain (NICD) which translocates to the nucleus and activates the expression of Notch target genes (Bray, 2006). In addition, Notch signaling activation may occur through non-canonical means on the endosomal surface, with or without stimulation by ligands (Tagami et al, 2008; Hori et al, 2012).

As a membrane-tethered transcription factor, the Notch signaling output is not only controlled by ligands, ADAM metalloproteases, the γ-secretase complex, and the CBF1, Suppressor of Hairless, Lag-1 (CSL) transcriptional complex but also through intracellular trafficking (Sakata et al, 2004; Bray, 2006; Fortini & Bilder, 2009; Andersson et al, 2011; Moretti & Brou, 2013; Schneider et al, 2013). Indeed, loss or modulation of various activities occurring in membranes positively or negatively impact signaling in both model organisms and humans. These activities include glycosylation enzymes (Panin et al, 1997; Shi & Stanley, 2003; Acar et al, 2008), internalization regulators such as dynamin (Seugnet et al, 1997),

---

[1]Dipartimento di Bioscienze, Università degli Studi di Milano, Milano, Italy   [2]Center for Genomic Science of IIT@SEMM, Fondazione Istituto Italiano di Tecnologia (IIT), Milan, Italy

Correspondence: thomas.vaccari@unimi.it; fkobia@associates.mku.ac.ke
Francis M Kobia's present address is Mount Kenya University, Department of Research and Innovation, Thika, Kenya
Mark Wade's present address is Astex Pharmaceuticals, Cambridge, UK

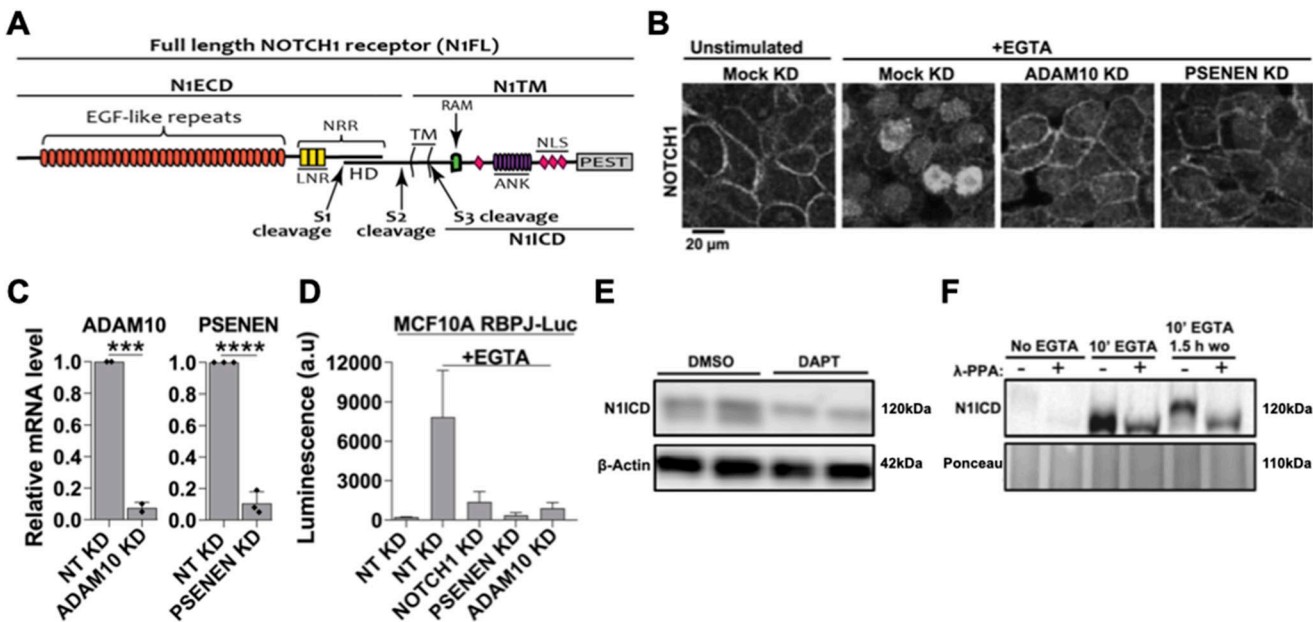

**Figure 1. Manipulation of endogenous NOTCH1 signaling in MCF10A cells.**
**(A)** The NOTCH1 (N1) full length receptor (N1FL) is composed of the N1 extracellular domain (N1ECD) and of the N1 transmembrane domain (N1TM). The N1ECD contains several EGF-like repeats, some of which are binding sites for Notch ligands and the negative regulatory region, which includes the Lin12/Notch Repeats (LNR) and the heterodimerization domain (HD). The N1ECD and N1TM portions of Notch receptors are produced by Furin cleavage occurring at the S1 site during trafficking to the trans-Golgi network and are non-covalently held together by Ca2+ at the HD. Ca2+ depletion causes ADAM10 and γ-secretase to sequentially cleave the receptor at S2 and S3 sites. This leads to the release in the cytoplasm of the N1 intracellular domain (N1ICD), which translocates to the nucleus. N1ICD contains the RBPJ-kappa-associated module, ankyrin repeats (ANK), nuclear localization signals, and the proline (P), glutamic acid (E), serine (S), and threonine (T) – PEST domain, which limits NICD half-life. **(B)** Immunofluorescence reveals that EGTA treatment relocates most endogenous NOTCH1 from the cell surface to the nucleus and this is inhibited by silencing ADAM10 or PSENEN. **(C)** siRNA against ADAM10 or PSENEN effectively depletes their mRNA. **(D)** The Notch reporter cell line, MCF10A-RbpJk-Luc, reports strong EGTA-induced Notch signaling, and this effect is markedly suppressed by silencing NOTCH1, PSENEN, or ADAM10. **(E)** Western blot analysis of MCF10A cell extracts using an antibody against S3-cleaved N1 shows that unstimulated cells express low levels of N1ICD, which appears as a doublet. γ-secretase inhibition for 3 h (DAPT) markedly reduces basal levels of the N1ICD lower band. **(F)** Western blot analysis of MCF10A lysates indicates that λ-phosphatase (λ-PPA) treatment leads to disappearance of the upper band of the N1ICD doublet, indicating that it corresponds to a phosphorylated N1ICD form. *** and **** indicate $P < 0.001$ and $P < 0.0001$, respectively.

and ubiquitin ligases such as Dx (Deltex) (Diederich et al, 1994; Hori et al, 2004; Kishi et al, 2001; Matsuno et al, 1995; Xu & Artavanis-Tsakonas, 1990), Su(Dx) (suppressor of Deltex/AIP4/ITCH in mammals) (Cornell et al, 1999; Chastagner et al, 2008), and NEDD4 (Sakata et al, 2004; Wilkin et al, 2004). In addition, components of the endo-lysosomal system, such as Endosomal Sorting Required for Transport (ESCRT) proteins (Thompson et al, 2005; Vaccari & Bilder, 2005; Hori et al, 2011) or the vacuolar-ATPase (V-ATPase) pump are required for endocytic Notch trafficking and signaling (Yan et al, 2009; Sethi et al, 2010; Vaccari et al, 2010; Lange et al, 2011; Kozik et al, 2013; Valapala et al, 2013; Kobia et al, 2014). It is plausible that numerous other factors, yet to be identified and characterized, contribute to fine tune Notch liberation from membrane signaling by harnessing the intracellular trafficking machinery. Thus, aiming to improve knowledge of Notch receptor trafficking and to identify novel genes that control Notch trafficking and signaling, here we combine quantitative immunofluorescence, RNA interference, and automated microscopic imaging to describe the kinetics of endogenous NOTCH1 export to the plasma membrane (PM) and its endocytic uptake and lysosomal clearance in human epithelial mammary gland cells in culture. Finally, we screen to identify novel modulators of Notch trafficking and signaling.

# Results

## Canonical NOTCH1 signaling in human MCF10A cells

Knowledge about Notch signaling dynamics is a prerequisite to investigating factors that control signaling by modulating its trafficking to subcellular compartments. MCF10A cells, derived from non-transformed human breast epithelial cells (Puleo & Polyak, 2021), express Notch signaling pathway ligands and receptors including *NOTCH1*, *NOTCH2*, and *NOTCH3*, whereas *NOTCH4* is not detectable (Kobia et al, 2014). Immuno-localization with an antibody that recognizes the intracellular domain of NOTCH1 (Fig 1A), shows that endogenous NOTCH1 is localized on the cell surface of confluent MCF10A cells (Fig 1B). As expected, efficient knockdown (KD) of key components of the Notch signaling pathway, such as *ADAM10* (encoding the S2 cleavage metalloprotease) and *PSENEN* (encoding a γ-secretase component necessary for S3 cleavage) in MCF10A cells prevented the translocation of endogenous NOTCH1 into the nucleus upon EGTA-mediated Ca$^{2+}$ chelation (Fig 1B and C; see the Materials and Methods section), which mimics ligand-dependent activation (Rand et al, 2000). Consistent with this, *NOTCH1*, *ADAM10*, or *PSENEN* depletion significantly suppressed EGTA-stimulated Notch signaling, when measured in MCF10A cells

stably expressing the Notch activation reporter RBPj-luc (Fig 1D), indicating that signaling in MCF10A in large part depends on proteolytic processing of NOTCH1, rather than on paralogs.

We previously observed that the S3-cleaved NOTCH1 intracellular domain (N1ICD) is visible in western blots as a band doublet around the 120 kD region (Kobia et al, 2014) (Fig 1E). We suspected that the slightly higher molecular weight band of the doublet may represent the N1ICD that is phosphorylated and destined for proteasomal degradation, whereas the lower molecular weight band represents newly generated N1ICD that has not been phosphorylated (Fryer et al, 2004; Li et al, 2014; Morrugares et al, 2020). Indeed, in extracts of cells treated with the γ-secretase inhibitor DAPT to block the production of N1ICD, we observed the presence of the upper band only (Fig 1E). To test if the upper band of the N1ICD doublet was indeed phosphorylated, we harvested proteins from unstimulated MCF10A cells, or from cells stimulated with EGTA for 10 min to induce N1ICD generation, or from cells stimulated with EGTA for 10 min and then returned to normal medium for 1.5 h. We then treated the lysates with or without λ-phosphatase (λ-PPA) (Satinover et al, 2004). λ-PPA treatment resulted in a minor downshift of the N1ICD in extracts stimulated with EGTA for 10 min and in a large downshift in those in which the EGTA had been washed out for 1.5 h after stimulation, indicating that N1ICD is progressively phosphorylated shortly after S3 cleavage (Fig 1F). Together, these data indicate that MCF10A cells are endowed with an endogenous pool of PM-localized NOTCH1 that can be canonically activated in a S2 and S3-dependent fashion to promote target gene transcription before being phosphorylated.

## Dynamics of NOTCH1 receptor trafficking in MCF10A cells

Because NOTCH1 is the receptor mostly contributing to Notch signaling in MCF10A cells, we next studied its trafficking. To this end, we first treated MCF10A cells with the V-ATPase inhibitor BAfA1 for 4 h to block lysosomal protein degradation. We previously observed that BAfA1 treatment blocks lysosomal acidification and reduces Notch signaling activation in MCF10A cells (Kobia et al, 2014; Tognon et al, 2016). Whereas most of NOTCH1 is localized to the PM in control-treated cells, in BafA1-treated cells NOTCH1 accumulates in slightly expanded LAMP1-positive lysosomal compartments. A similar pattern of localization is observed for EGFR (Fig 2A, quantified in A'). These data indicate that, in unstimulated conditions, endogenous NOTCH1 and EGFR localizations represent the steady state of receptors that are continuously trafficked to lysosomes for degradation.

Because new synthesis of NOTCH1 is expected to compensate for its rapid lysosomal degradation, we studied NOTCH1 exocytic trafficking relative to EGFR. Consistent with continuous secretory trafficking of newly synthesized receptors, we observed that in untreated cells an intracellular pool of NOTCH1 and EGFR colocalizes with the ER marker RTN3 (reticulon-3; Fig 2B). We reasoned that Ca$^{2+}$ chelation, in addition to triggering shedding of the extracellular portion of Notch, when prolonged, is known to lead to exhaustion of the intracellular Ca$^{2+}$ stores and membrane fusion arrest (Pryor et al, 2000; Hay, 2007). Indeed, 4-h treatment with 2.5 mM EGTA results in clearing of the PM of existing NOTCH1

molecules and to accumulation of most NOTCH1 and EGFR in a RTN3-positive compartment (Fig 2B; note that EGFR is still visible at the cell surface). This experiment highlights a strategy for controlling the release of cargoes and tracking their trafficking to the PM. Thus, we immunolabeled cells to detect NOTCH1, EGFR, and Giantin, a marker of the Golgi apparatus (GA), to detect the intracellular pool of receptors that transits to the GA (Fig 2C).

To time endogenous NOTCH1 and EGFR trafficking from the ER to the cell surface, we washed out the EGTA-containing media after the 4-h treatment with fresh complete medium (w/o). Treating the cells with EGTA for 4 h caused accumulation of most NOTCH1, EGFR in the ER, and alteration of Giantin localization (Fig 2C, quantified in C'). Strikingly, both NOTCH1 and EGFR were almost exclusively localized in the GA upon 1 h w/o, indicating that NOTCH1 and EGFR trafficking is rapidly resumed after EGTA w/o. A 4-h w/o fully restored NOTCH1 localization to the GA, whereas EGFR localization to the GA did not normalize. In contrast, a 4-h w/o fully restored EGFR localization to the cell surface, whereas that of NOTCH1 required longer (Fig 2C, quantified in C'). These data reveal time-resolved secretion of NOTCH1 and EGFR from the ER to the GA, and to the PM, and indicate that the bulk of the NOTCH1 pool progresses from the ER to GA and from GA to the cell surface in about 4 h. Slight differences between trafficking of NOTCH1 and EGFR suggest that the two receptors might possess distinct trafficking and processing requirements. Our analyses of the timing of endogenous NOTCH1 and EGFR secretory trafficking are consistent with the timing of endocytic degradation obtained by limiting lysosomal activity with BafA1 treatment and suggest that in MCF10A cells NOTCH1 and EGFR molecules possess a limited lifespan of about 8 h.

## High-content screening identifies 51 novel potential modulators of intracellular NOTCH1 localization, 39 of which affect signaling

The limited lifespan of endogenous NOTCH1 and EGFR in MCF10A cells predicts that perturbation of genes that control their steady-state localization might significantly alter signaling. Thus, we tested whether the expression of 2,749 human genes with known or predicted small compound inhibitors belonging to different functional categories (see Supplemental Data 1) is required for correct NOTCH1 localization in MCF10A cells. To this end, we reverse transfected cells on 10 arrayed plates containing individual pools of 4 silencing RNAs (siRNA) targeting each gene and controls in unstimulated cells (Fig 3A). To identify genes that are also important for NOTCH1 nuclear localization upon signaling activation, we subjected duplicate plates to EGTA stimulation. We then stained cells with an antibody that recognizes the cytoplasmic portion of NOTCH1. The cells were counterstained with DAPI and phalloidin to visualize nuclei and cell cortices, respectively (Fig 3B). We devised an automated high-content image analysis pipeline to identify factors that, when silenced, affected NOTCH1 subcellular localization by altering receptor amounts on the cell surface, in the cytoplasm, or in the nucleus (Fig 3B). In such a primary screen, we identified 231 potential modulators of NOTCH1 localization (Fig 4A; Supplemental Data 2). We then retested the positive hits using a custom-made plate containing 231 individual esiRNA; a second, independent set of gene-silencing reagent comprised a heterogeneous mixture of numerous

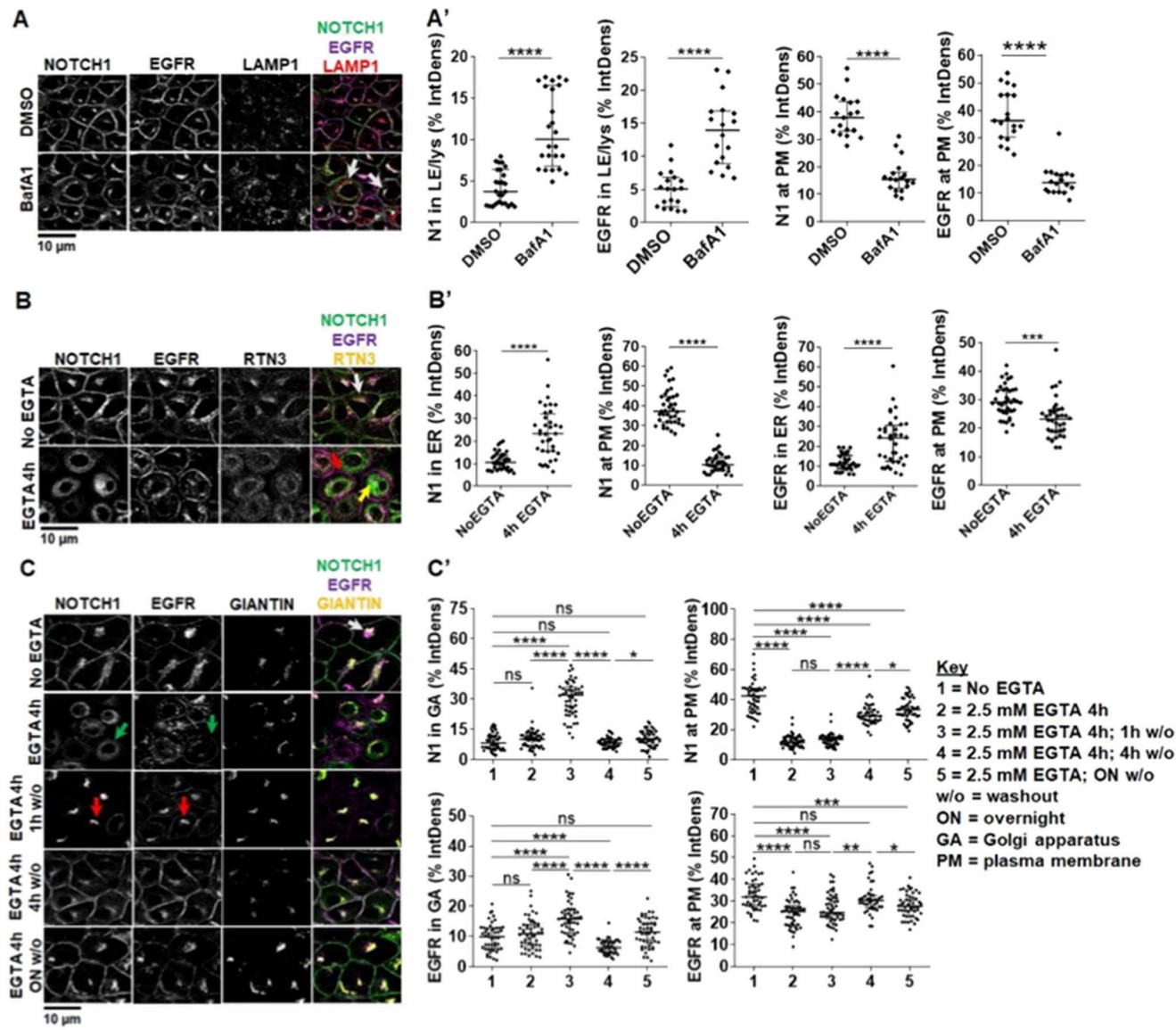

**Figure 2. Endocytic and exocytic endogenous NOTCH1 receptor trafficking in MCF10A cells.**
**(A)** Confocal sections of MCF10A cells treated and immunolabeled as indicated. Upon BafA1 treatment, plasma membrane (PM) localization of both NOTCH1 and EGFR is reduced, and both receptors accumulate in LAMP1-positive compartments (arrows) (A') Quantification of panel (A). **(B)** Confocal sections of MCF10A cells treated and immunolabeled as indicated. 4-h EGTA treatment depletes NOTCH1 from the PM and strongly accumulates it in a RTN3-positive compartment (yellow arrow), whereas EGFR remains on the PM (red arrow). In unstimulated cells, a pool of both NOTCH1 and EGFR is visible in the RTN3-positive compartment (white arrow). **(B')** Quantification of panel B. **(C)** Confocal sections of MCF10A cells treated and immunolabeled as indicated. An intracellular pool of NOTCH1 and EGFR colocalizes with the Golgi apparatus (GA) marker, GIANTIN (white arrow). Compared with non-EGTA–treated cells, 4-h EGTA treatment causes diffused NOTCH1 and EGFR accumulation in the cytosol, with some EGFR remaining on the PM (green arrows). The GIANTIN signal is also diffused consistent with an expected fragmentation of the GA. EGTA washout (w/o) for 1 h causes all NOTCH1 and most EGFR signal to localize in the GA (red arrows). EGTA w/o for 4 h or overnight (ON) restores normal intracellular distribution of NOTCH1 and EGFR. **(C')** Quantification of panel (C). *, **, ***, ****, ns indicate $P < 0.05$, $P < 0.01$, $P < 0.001$, $P < 0.0001$, and not significant, respectively).

siRNA targeting each gene (Theis & Buchholz, 2010) identified on the primary screen. This validation screen yielded 73 hits (Fig 4B). Only 51 of them confirmed the effects observed in the primary screen. Of these, the majority (44/51) increased the intracellular pool of NOTCH1 either in unstimulated (NoEGTA, 36 hits) or in stimulated conditions (+EGTA, 8 hits), whereas a minority altered both PM and intracellular NOTCH1 or increased nuclear NOTCH1 (Fig 4C; Supplemental Data 3). Representative examples of hits belonging to each phenotypic category are shown is Fig S1.

To systematically determine the impact of the potential modulators of NOTCH1 trafficking on Notch signaling, we individually silenced with esiRNA the 231 primary hits also in MCF10A-RbpJk-Luc cells with or without EGTA stimulation and measured signaling output (Supplemental Data 2). Focusing on the validated 51 genes that affect localization in the secondary screen, the analysis identified 39 depletions (76.5% of the total) that altered Notch signaling by at least 30%, with 21 modulating signaling in unstimulated conditions only, 8 in stimulated conditions only, and 10 in

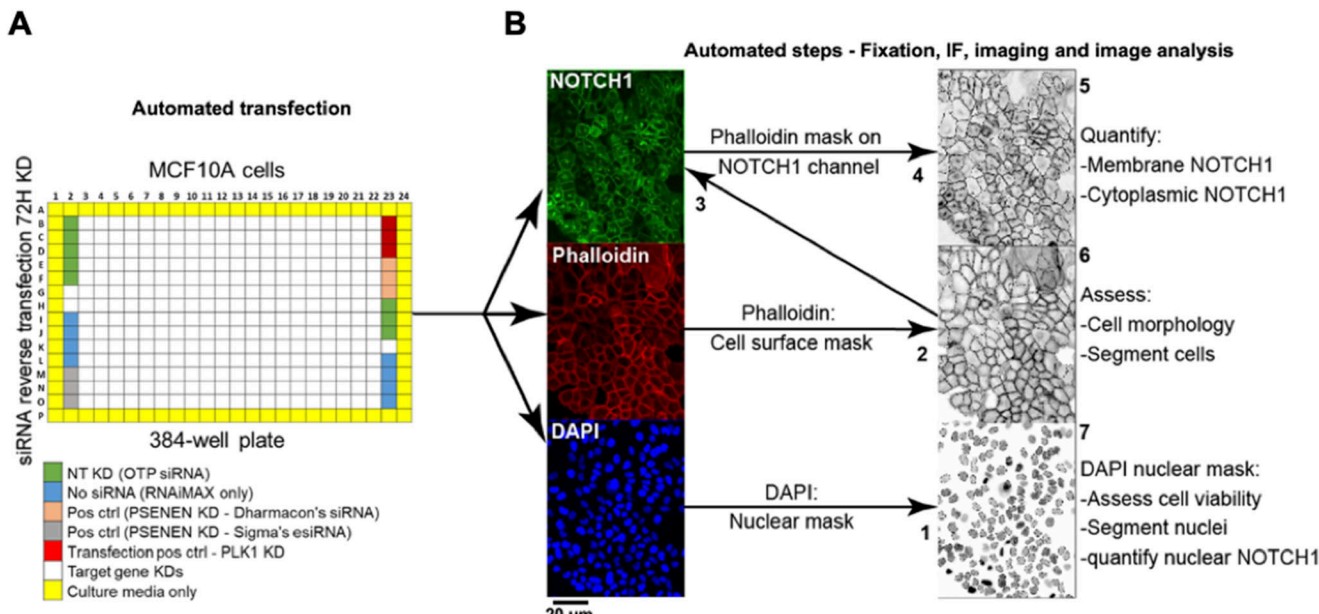

**Figure 3. High-content screen pipeline and image analysis strategy.**
**(A)** The primary screen was performed on MCF10A cells in a 384-well format. Genes belonging to a subset of the human genome were knocked down in six replicate plates for 72 h by RNAi along with the listed controls. In a duplicate experiment, cells were stimulated with EGTA. Control wells were distributed on the plates as shown. NT: non-targeting KD – cells were transfected with non-targeting siRNA. No siRNA: cells received RNAiMAX only. Positive control (pos ctrl) – cells were transfected with siRNA against PSENEN. Transfection pos ctrl: cells were transfected with siRNA against polo-like kinase 1 (PLK1) as a cytotoxic indicator of transfection efficiency. Outer wells (yellow) were filled with media to prevent evaporation in experimental wells. **(B)** To assess endogenous NOTCH1 localization, cells in plates were subjected to automated IF and DAPI/phalloidin labeling to demarcate nuclei and the cell cortex, respectively. The script steps of the image acquisition and analysis developed in-house are as follows: (1) The nuclei were segmented based on DAPI signal. (2) The cell surface was segmented using the phalloidin signal and overlaid on the NOTCH1 channel (3) to quantify levels of cell surface NOTCH1 (4), and the area between the cell cortex and the nuclei was used to determine levels of cytoplasmic NOTCH1 (5). (6) The phalloidin mask was also used to count cells and establish cell-to-cell boundaries. (7) Nuclear size was used as readout of cell viability as compact; pyknotic nuclei identify dead cells. Using such a pipeline, each gene KD was defined by its effects on the amount of NOTCH1 in the cell cortex, in the cytoplasm or in the nucleus.

both conditions (Fig 4D and E; Supplemental Data 3). Of the 39 depletions, 30 led to reduced signaling (nine of which also in stimulated conditions), whereas nine led to increased signaling (eight of which in stimulated conditions) (Fig 4F). The NOTCH1 localization phenotypic classes of the 39 depletions are reported in Fig 4G. This combined approach led to the identification of several new positive and negative regulators of Notch trafficking and signaling that require further characterization.

## Modulation of NOTCH1 localization and signaling by *PTPN23*

Considering the role of endocytosis in the regulation of Notch localization, to validate high-content screening (HCS) hits, we began from *PTPN23*, one of the eight identified genes that have been previously associated with regulation of endo-lysosomal trafficking (16% of the confirmed 51 hits: *CPA2, CTSE, MLC1, PIK3C2G, PTPN23, TINAGL1, ZNRF1, ZNRF2*). *PTPN23* encodes HD-PTP, a ubiquitous non-receptor tyrosine pseudophosphatase that has been involved extensively in endosomal sorting, multivesicular body formation and ESCRT activity (Doyotte et al, 2008; Tabernero & Woodman, 2018). Automated analysis of HCS IF images revealed that relative to control cells in which NOTCH1 mainly resided at the PM, *PTPN23* KD MCF10A cells exhibited marked intracellular NOTCH1 accumulation, which appeared as puncta (Fig S2A). We reasoned

that *PTPN23* depletion might interfere with NOTCH1 endosomal trafficking or sorting. To identify the compartment in which the NOTCH1 accumulated, we co-stained cells with antibodies against NOTCH1 and the endo-lysosomal markers EEA1 or LAMP1, which identify early endosomes and lysosomes, respectively. Efficient depletion of *PTPN23* (Fig S2B) led to significant expansion of the cytoplasmic pool of puncta positive for EEA1, LAMP1, and NOTCH1 (Fig 5A, quantified in A'). Quantitative analysis also revealed that the NOTCH1 that accumulated upon *PTPN23* depletion mainly resided in puncta positive for EEA1 (Fig 5A'). Taken together, these data indicated that *PTPN23* depletion might interfere with early to late endosomal trafficking, thereby trapping NOTCH1 mostly in early endosomes.

We next evaluated how loss of *PTPN23* might affect Notch processing and signaling. Western blot analysis revealed a marked reduction in both N1FL and N1TM upon *PTPN23* KD (Fig 5B). Because MCF10A cells also express NOTCH2 and 3, we repeated the analysis using antibodies to detect N2FL and N2TM as well as N3FL and N3TM. A similar reduction to that of N1FL and N1TM was observed for N2FL and N2TM upon *PTPN23* KD. In contrast, levels of N3FL or N3TM were not significantly reduced (Fig S3A and B). Whereas it is unclear why the NOTCH3 might differ from other Notch paralogs, we note that NOTCH3 expression in MCF10A is very limited when compared with NOTCH1 and 2 (Fig S4). We next assessed N1ICD levels. Surprisingly,

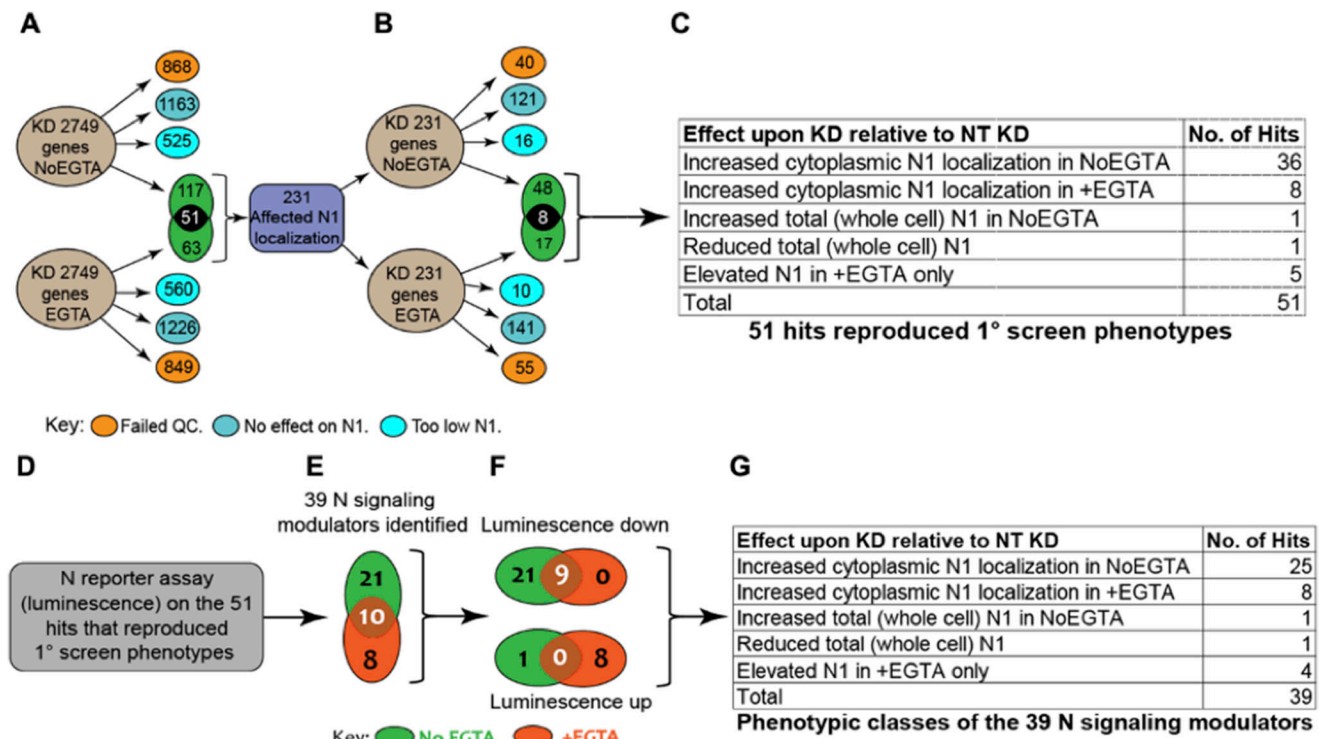

**Figure 4. Schematics of the screens and of the candidate gene classification process.**
**(A)** A total of 2,749 genes were silenced in the primary screen. Candidates that, when compared with controls, led to marked cytotoxicity, did not affect intracellular NOTCH1 (N1) localization, or caused general loss of NOTCH1 signal, were excluded from further analysis (orange, blue, and light blue circles, respectively). A total of 231 genes that altered intracellular NOTCH1 localization were identified (green). 117 did so in unstimulated condition, 63 upon EGTA stimulation and 51 in both conditions. **(B)** The 231 candidates underwent secondary screening. 73 of the 231 led to changes in NOTCH1 localization, but only 51 genes reproduced the primary screen phenotypes. **(C)** Of these 51, 38 affected NOTCH1 trafficking in the unstimulated No EGTA condition, 13 altered NOTCH1 trafficking in the stimulated +EGTA condition. **(D, E)** MCF10A-RbpJk-Luc Notch reporter assay identifies 39 candidates that affect Notch signaling. **(F)** Of the 39, 30 suppress Notch signaling, whereas nine enhance Notch signaling. **(G)** Classification of the 39 candidates by their effect on intracellular NOTCH1 (N1) localization and/or levels.

these were not significantly altered, both in unstimulated and stimulated conditions (Fig 5C). To assess the effect of *PTPN23* depletion on Notch signaling, we measured Notch-mediated RBPj-luc expression and found that when compared with control cells, *PTPN23* silencing significantly suppressed Notch signaling in unstimulated conditions but not in the presence of EGTA (Fig 5D). Similar observations were obtained when measuring *HES1* and *HEY1* expressions by quantitative reverse transcription PCR (RT-qPCR) (Fig 5E and F). Because reduced biosynthetic N1FL levels could also result from low *NOTCH1* transcription, we also measured *NOTCH1* mRNA levels and found that upon *PTPN23* depletion, *NOTCH1* expression levels were mildly but significantly reduced (Fig 5G). These data indicate that *PTPN23* is required for NOTCH1 sorting into multivesicular bodies and for sustaining Notch expression and basal signaling and illustrate the screen's ability to identify novel modulators of NOTCH1 trafficking and signaling.

### HCN2 silencing traps NOTCH1 in an enlarged GA and suppresses Notch signaling

Automated analysis of HCS IF images identified four gene encoding channels (*CACNB4*, *TRPM7*, *HCN2*, and *MLC1*) that upon KD in unstimulated conditions, caused cytosolic NOTCH1 accumulation

(of a total of 44 hits, a 9.1% enrichment; Supplemental Data 3). Of these, *HCN2* KD led to the most striking phenotype, with strong NOTCH1 accumulation in the perinuclear space when compared with mock-silenced (non-targeting/NT) controls (Fig S5A). To test if NOTCH1 accumulated in the perinuclear GA, we knocked down *HCN2* (Fig S5B) and co-stained for NOTCH1 and the cis-GA protein Giantin. Confocal analysis confirmed the perinuclear NOTCH1 localization observed in the HCS images and revealed an expansion of the Giantin-positive GA and increased localization of NOTCH1 in the cis-GA compartment, when compared with control cells (Fig 6A, quantified in A'). In addition, the analysis revealed that when compared with the control, *HCN2* silencing markedly reduced cell surface NOTCH1 levels (Fig 6A, quantified in A'), suggesting that HCN2 silencing might interfere with trafficking of newly made NOTCH1 to the cell surface. To assess this, we compared NOTCH1 and EGFR localization upon *HCN2* KD and 4-h EGTA treatment. Both NOTCH1 and EGFR mostly failed to return to the PM after 1 or 4-h w/o (Fig S6A; quantified in A'), indicating that both receptors do not traffic efficiently to the PM in the absence of HCN2. Relative to mock silencing, *HCN2* depletion also led to alteration of the TGN with loss of TGN-associated marker TGN46 and slight elevation of TGN-associated marker golgin-97 (Fig 6B, quantified in B').

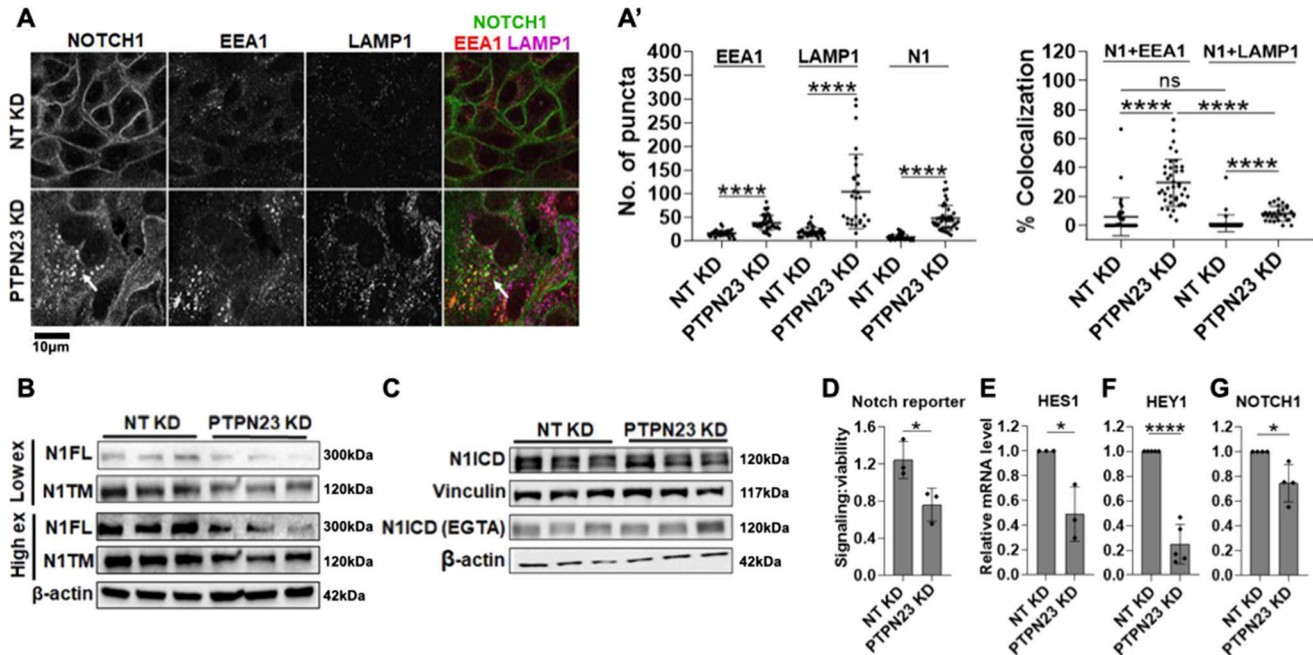

**Figure 5. Analysis of NOTCH1 localization and signaling activity upon PTPN23 silencing.**
**(A)** Confocal sections of MCF10A cells treated and immunolabeled as indicated. **(A')** PTPN23 silencing causes marked increase in the number of EEA1, LAMP1, and NOTCH1 (N1) puncta, as well as accumulation of intracellular NOTCH1-positive puncta, which mostly colocalize with EEA1 (arrows; quantified in (A')). **(B, C)** Western blot analyses indicate that PTPN23 depletion reduces the levels of N1FL and N1TM but does not alter N1ICD levels with or without EGTA. **(C)** Note that levels of N1ICD in unstimulated and EGTA-stimulated cells are not comparable in (C) because of different detection methods (see the Materials and Methods section). **(D, E, F, G)** The Notch reporter cell line MCF10A-RbpJk-Luc and RT-qPCR analysis reveal that PTPN23 depletion suppresses basal Notch signaling. *, ****, and ns indicates $P < 0.05$, $P < 0.0001$, and not significant, respectively.

Next, we extended the analysis of *HCN2* KD to Notch signaling output. *HCN2* depletion significantly reduced expression of a Notch signaling reporter in unstimulated cells (Fig 6C) and limited the expression of the NOTCH1 target genes, *HES1* and *HEY1*, relative to controls (Fig 6D and E). Consistent with these observations, Western blot analysis revealed that when compared with the control, *HCN2* silencing markedly reduced the levels of N1ICD both in the absence and presence of EGTA (Fig 6F). Intriguingly, the levels of the S1 substrate, N1FL, revealed the opposite trend, with *HCN2* KD elevating N1FL levels relative to the control (Figs 6G and S5D). In contrast, the levels of N1TM, the S2-processed form of NOTCH1, were reduced in *HCN2* KD cells, when compared with controls (Fig 6G). Similar results were obtained by analyzing NOTCH2 and NOTCH3 levels (Fig S3A and B). Together, these findings suggest that HCN2 is required to support a TGN organization that allows S1 processing, PM delivery and signaling of NOTCH1-3 receptors.

## SGK3 silencing elevates cytosolic NOTCH1 levels

Automated image analysis identified five kinases that, when silenced, interfered with NOTCH1 localization *DCK, EIF2AK1, MASTL, PIK3C2G, SGK3* (corresponding to 9.8% of 51 hits; Supplemental Data 3). Of these, *SGK3* might regulate Notch receptor stability based on previous findings that SGK3 targets NDRG1 for FBW7-mediated proteasomal degradation (Gasser et al, 2014), a process that also limits N1ICD stability (Öberg et al, 2001). Moreover, the SGK3 paralog, SGK1, has previously been reported to negatively modulate Notch signaling by targeting NOTCH1 for proteasomal degradation (Mo

et al, 2011). Automated HCS image analysis revealed that relative to mock silencing, efficient *SGK3* KD causes marked NOTCH1 accumulation in the cytosol (Fig S7A). To validate this observation, we silenced *SGK3* (Fig S7B and C) and analyzed the cells by confocal microscopy after co-staining with antibodies against golgin-97, EEA1 and LAMP1, to mark the early TGN, early endosomes, and lysosomes, respectively. This analysis confirmed the presence of elevated intracellular NOTCH1 levels but revealed no significant difference in its colocalization with golgin-97-, EEA1-, and LAMP1-positive compartments (Fig 7A and B, quantified in A'-B'). We also observed a significant elevation of EEA1 and LAMP1 puncta, suggesting that SGK3 might regulate the endocytic compartment.

Given the diffused accumulation of NOTCH1 observed upon *SGK3* silencing, we wondered whether this correlated with elevated NOTCH1 processing and signaling levels. We first tested which NOTCH1 form accumulated upon *SGK3* KD and found that *SGK3* depletion leads to elevated N1FL, N1TM, and N1ICD levels in both unstimulated and stimulated conditions (Fig 7C and D). When we analyzed NOTCH2 and NOTCH3 levels, we observed elevated NOTCH3 but not NOTCH2 levels (Fig S3A and B). We next used the Notch signaling reporter cell line MCF10A-RbpJk-Luc and observed that when compared with the control, silencing *SGK3* caused significantly higher Notch signaling output upon EGTA stimulation (Fig 7E). Although signaling did not differ significantly in unstimulated *SGK3*-silenced MCF10A-RbpJk-Luc cells when compared with mock silencing, RT-qPCR analysis revealed that *SGK3* KD in MCF10A cells significantly up-regulated the expression of the Notch signaling targets *HEY1*, but not HES1 in unstimulated conditions. Depletion

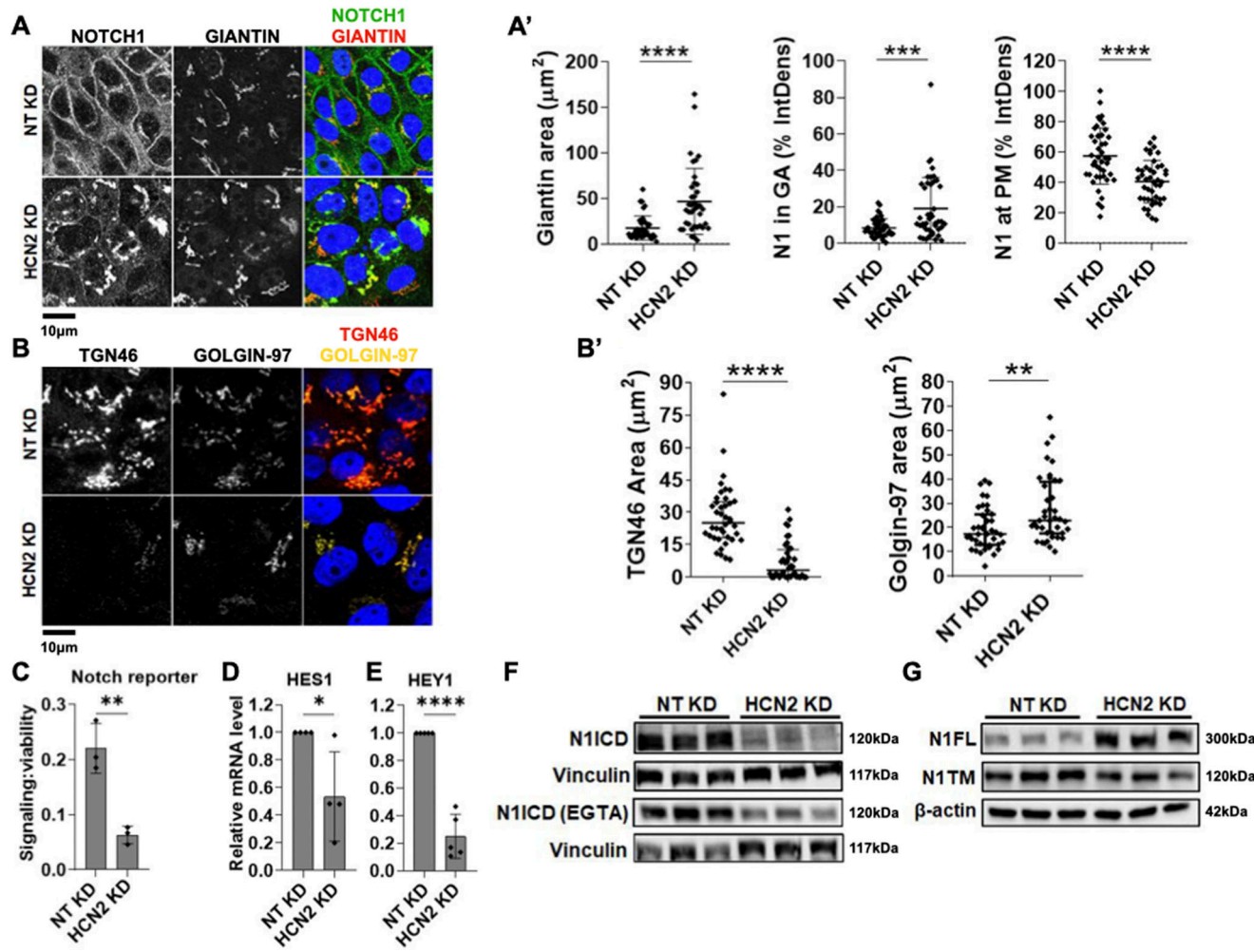

**Figure 6. Analysis of NOTCH1 localization and signaling activity upon HCN2 silencing.**
**(A)** Confocal sections of MCF10A cells treated and immunolabeled as indicated. **(A')** HCN2 silencing causes marked NOTCH1 accumulation in the Giantin-positive cis-GA compartment when significantly reducing cell surface NOTCH1 (N1) levels (quantified in (A')). **(B)** Confocal sections of MCF10A cells treated and immunolabeled as indicated. **(B')** HCN2 silencing markedly depletes the TGN46-positive TGN compartment but not the golgin-97-positive TGN compartment (quantified in (B')). **(C, D, E)** The Notch reporter cell line, MCF10A-RbpJk-Luc and RT-qPCR analysis reveal that HCN2 silencing suppresses basal Notch signaling. **(F)** Western blot analyses indicate that HCN2 depletion reduces both basal and EGTA-stimulated N1ICD levels. **(F)** Note that levels of N1ICD in unstimulated and EGTA-stimulated cells are not comparable in (F) because of different detection methods (see materials and methods). **(G)** Western blot analysis indicates that HCN2 silencing results in marked accumulation of N1FL. *, **, ****, and ns indicate $P < 0.05$, $P < 0.01$, $P < 0.0001$, and not significant, respectively.

led also to a transcriptional elevation of NOTCH1 both in unstimulated and EGTA-stimulated cells (Fig 7F–H).

The observation that depleting *SGK3* in MCF10A cells leads to elevated N1ICD levels made us wonder whether SGK3 reduction might impair N1ICD degradation. To test this possibility, we silenced *SGK3* and then treated the cells with cycloheximide (CHX) to block protein synthesis or with DMSO as a negative control. Because blocking the synthesis of new proteins would allow cells to turn over proteins that had been present before CHX addition, we reasoned that if SGK3 is required for efficient N1ICD degradation, upon CHX treatment *SGK3*-depleted cells would exhibit elevated N1ICD levels when compared with control cells. Western blot analysis using antibodies against N1FL and N1ICD revealed that *SGK3* silencing caused a marked increase in the levels of N1FL and N1ICD when compared with control. However, N1FL and N1ICD levels were markedly decreased in both control and *SGK3* KD cells upon CHX treatment (Fig S7D), indicating

that SGK3 is not required for N1ICD destabilization. Finally, to assess whether SGK3 contributed directly or indirectly to N1ICD phosphorylation, we silenced *SGK3* in MCF10A cells and then performed the λ-PPA assay. This analysis revealed that in *SGK3* KD cells, accumulated N1ICD was still phosphorylated, and that phosphorylation could be erased by treating the cell extracts with λ-PPA (Fig S7E), indicating that SGK3 is not required for N1ICD phosphorylation. Taken together, these data indicate that SGK3 negatively regulates NOTCH1 levels and signaling indirectly by a mechanism other than control of N1ICD stability and/or degradation.

## Pharmacologic SGK3 inhibition elevates N1ICD levels and Notch signaling

To explore whether SGK3 modulation of NOTCH1 could be controlled pharmacologically, we examined the effect of VPS34-IN1, a well

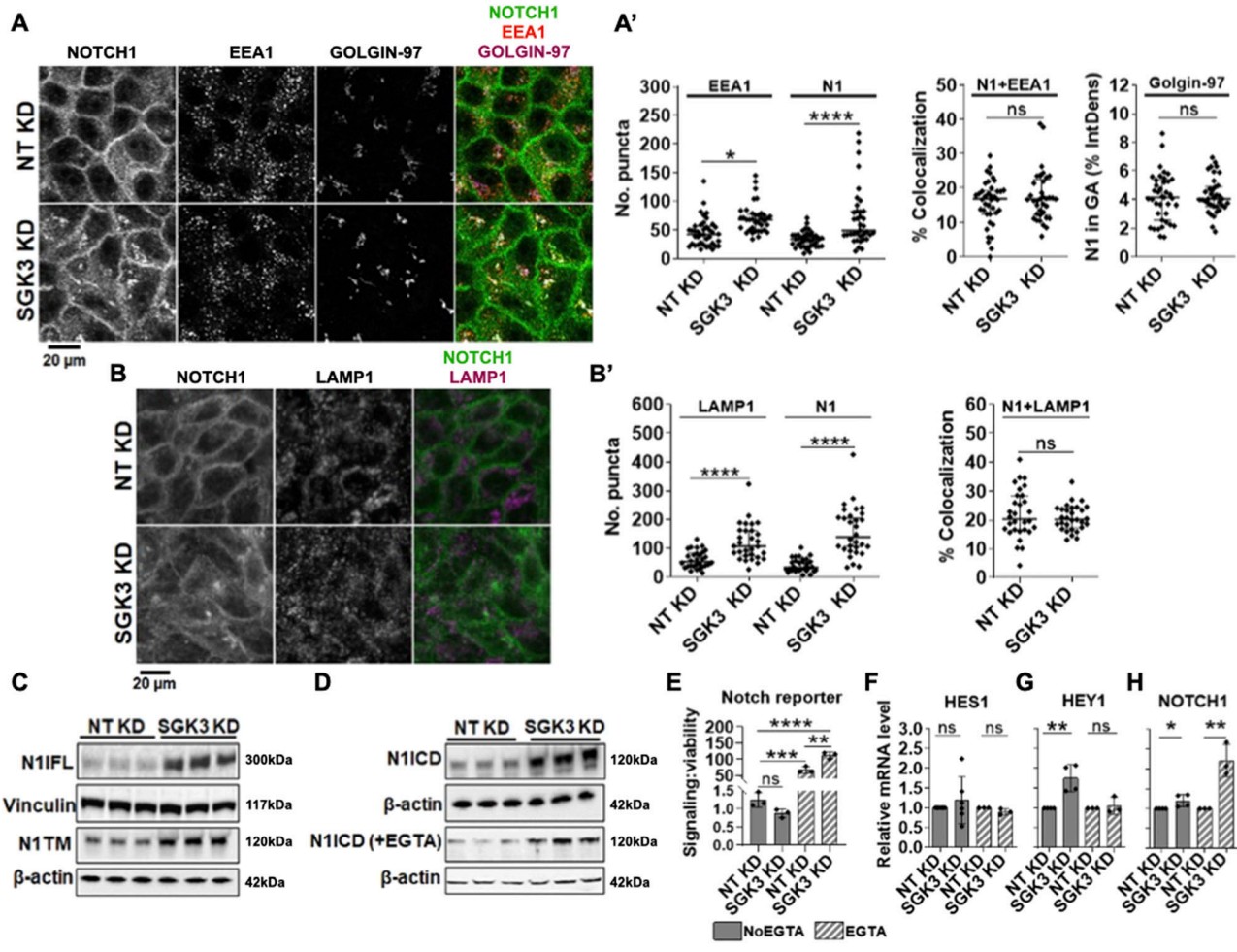

**Figure 7. Analysis of NOTCH1 localization and signaling activity upon SGK3 silencing.**
**(A)** Confocal sections of MCF10A cells treated and immunolabeled as indicated. **(A')** SGK3 silencing causes NOTCH1 (N1) accumulation in the cytoplasm without apparent colocalization with EEA1 or golgin-97 (quantified in (A')). **(B)** Confocal sections of MCF10A cells treated and immunolabeled as indicated. **(B')** The accumulated N1 upon SGK3 silencing does not colocalize with LAMP1 (quantified in (B')). **(C, D)** Western blot analyses show that SGK3 depletion causes strong accumulation of all forms of the NOTCH1 receptor. **(D)** Note that levels of N1ICD in unstimulated and EGTA-stimulated cells are not comparable in (D) because of different detection methods (see the Materials and Methods section). **(E)** The Notch reporter cell line MCF10A-RbpJk-Luc reveals that SGK3 silencing enhances Notch signaling in EGTA-stimulated cells. **(F, G, H)** RT-qPCR analysis reveals that SGK3 depletion enhances basal expression of NOTCH1 both unstimulated and EGTA-stimulated cells.

characterized SGK3 inhibitor (Bago et al, 2014; Ronan et al, 2014). We first tested if VPS34-IN1 could effectively block SGK3 activity in MCF10A cells. Because we were unable to directly evaluate SGK3 phosphorylation upon treatment of cultured cells with VPS34-IN1, we evaluated the phosphorylation levels of NDRG1, a known SGK3 substrate (Gasser et al, 2014; Tovell et al, 2019). SGK3 can be activated by stimulating the cells with growth factors, such as IGF1 and EGF (Kobayashi et al, 1999; Virbasius et al, 2001; Malik et al, 2018). Thus, we stimulated control or VPS34-IN1-treated cells with EGF to promote NDRG1 phosphorylation. Strikingly, EGF failed to elicit NDRG1 phosphorylation in VPS34-IN1-treated cells and erased the basal low level of NDRG1 phosphorylation in unstimulated cells (Fig 8A), indicating that treatment with VPS34-IN1 potently blocks SGK3 activity in cultured MCF10A cells.

Next, to determine if pharmacologic VPS34-IN1 treatment reproduces the effects of *SGK3* silencing on NOTCH1, we treated MCF10A cells with VPS34-IN1 and measured N1ICD levels. Western blot analysis

revealed that relative to mock treatment, VPS34-IN1 significantly elevated the levels of N1ICD (Fig 8B). To establish whether VPS34-IN1 treatment also elevates Notch targets, we examined the expression levels of *HEY1* by RT-qPCR. This analysis revealed that when compared with control cells, *HEY1* levels were significantly higher in VPS34-IN1-treated cells (Fig 8C). A similar increase in N1ICD levels was obtained by treating cells with SGK3-PROTAC, a small molecule compound inducing SGK3 degradation (Tovell et al, 2019) (Fig 8D, quantified in C'). Taken together, these data indicate that as with *SGK3* silencing, pharmacologic VPS34-IN1 or SGK3-PROTAC treatment elevates NOTCH1 levels and might promote Notch signaling.

### Depletion of novel Notch modulators in cancer cells

To test whether *PTPN23*, *HCN2*, or *SGK3* affect Notch levels in breast cancer cells, we first evaluated by Western blot the levels of NOTCH1-3

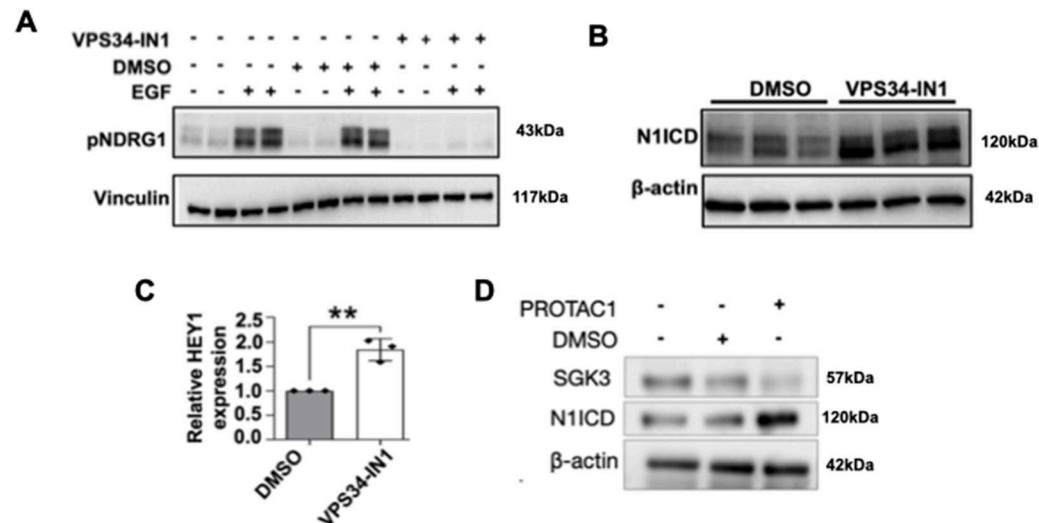

**Figure 8. Pharmacologic SGK3 inhibition elevates γ-secretase-derived N1ICD and Notch signaling.**
**(A)** Western blot analysis reveals that serum-starved, VPS34-IN1-treated, MCF10A cells fail to phosphorylate the SGK3 target NDRG1 upon EGF treatment. **(B)** Western blot analysis reveals that pharmacologic VPS34-IN1 treatment enhances N1ICD levels. **(C)** RT-qPCR analysis shows that pharmacologic VPS34-IN1 treatment significantly enhances the expression of the Notch target gene HEY1. **(D)** Western blot analysis confirms that a specific pharmacological treatment with PROTAC1 enhances N1ICD levels. ** indicates $P < 0.01$.

in a panel of five breast cancer lines of different origin. By comparing it with MCF10A, which express relatively high levels of all three Notch paralogs, we found that these express varying levels of Notch paralogs and produce different basal levels of N1ICD (Fig S4). Because MDA-MB-231 and BT-474 do not express high levels of NOTCH1 or paralogs and do not produce much N1ICD, we decided to exclude them from further experiments. We proceeded to evaluate MCF7 cells, which express low NOTCH1 and high levels of NOTCH2-3 but little basal N1ICD, and CAL-51 and MCF10DCIS.COM (Miller et al, 2000), which express levels of NOTCH1 and N1ICD comparable with those of MCF10A cells. We then evaluated levels of the NOTCH paralogs in each of these cell lines upon depletion of *PTPN23*, *HCN2*, or *SGK3* (Fig 9). We found that in most cell lines and for most Notch paralogs *HCN2* depletion led to an elevation of Notch FL levels, or of the Notch FL/TM ratio, while *PTPN23* depletion leads to their decrease. Finally, *SGK3* depletion caused an elevation of Notch and N1ICD levels but not in CAL-51 cells (Miller et al, 2000) (Fig 9A–C). Because *SGK3* appears to negatively regulate NOTCH1 levels, we depleted it in SCC022 skin cancer cells in which low levels of NOTCH1 promote tumorigenesis (Lefort et al, 2007). We observed that *SGK3* depletion resulted in elevated N1 TM, N1ICD levels, indicating that SGK3 contributes to the destabilization of NOTCH1 in skin cancer cells (Fig 9D). Overall, these data suggest that the perturbations of Notch paralogs observed in MCF10A cells might extend to transformed cells.

## Discussion

NOTCH1 activity in MCF10A cells has been mostly characterized using transfection of activated forms (Meurette et al, 2009; Mazzone et al, 2010). Here, we have shown that in MCF10A cells

is possible to activate endogenous NOTCH1 using a short pulse of $Ca^{2+}$ chelation that destabilizes the NOTCH1 HD domain leading to S2 and S3 cleavage, as previously reported for a variety of different cell types (Rand et al, 2000; Westhoff et al, 2009; Tremblay et al, 2013; Swaminathan et al, 2022). Interestingly, we find that confluent MCF10A cells possess a low level basal S3 cleavage and that, upon S3 cleavage NOTCH1 is rapidly phosphorylated, an event that has been described to lead to proteasomal degradation of NICD. Whether such basal S3 cleavage depends on transactivation among neighboring cells, considering that MCF10A cells also express NOTCH1 ligands (Kobia et al, 2014), or whether it is a cell autonomous occurrence because of intracellular ligand-independent activation remains to be determined.

By means that inhibit lysosomal degradation, we also have determined that trafficking of endogenous NOTCH1 in MCF10A cells occurs similarly to what has been reported in vivo in epithelial imaginal discs of *Drosophila melanogaster* (Vaccari et al, 2008). Interestingly, upon inhibition of lysosomal degradation, endogenous NOTCH1 and EGFR not only accumulate in lysosomes but are also depleted from the PM, suggesting that some coordination of new synthesis and/or ER to PM trafficking with clearance might exist. Transit assessment at different time points indicates that NOTCH1 traffics from the ER to the GA in less than 1 h and from the GA to PM in as little as 3 h. These times suggest that on average the half-life of each NOTCH1 molecule is only ~4 h, a time comparable with previous in vivo findings using *Drosophila* Notch (Couturier et al, 2014). Such half-life is also compatible with the two main events described during NOTCH1 secretion, namely receptor glycosylation and S1 cleavage by Furin (Logeat et al, 1998; Sasamura et al, 2007).

Because the half-life of NOTCH1 is very limited, we posited that genetic perturbations could result in visible changes in NOTCH1 localization. Such possibility suggested that high throughput

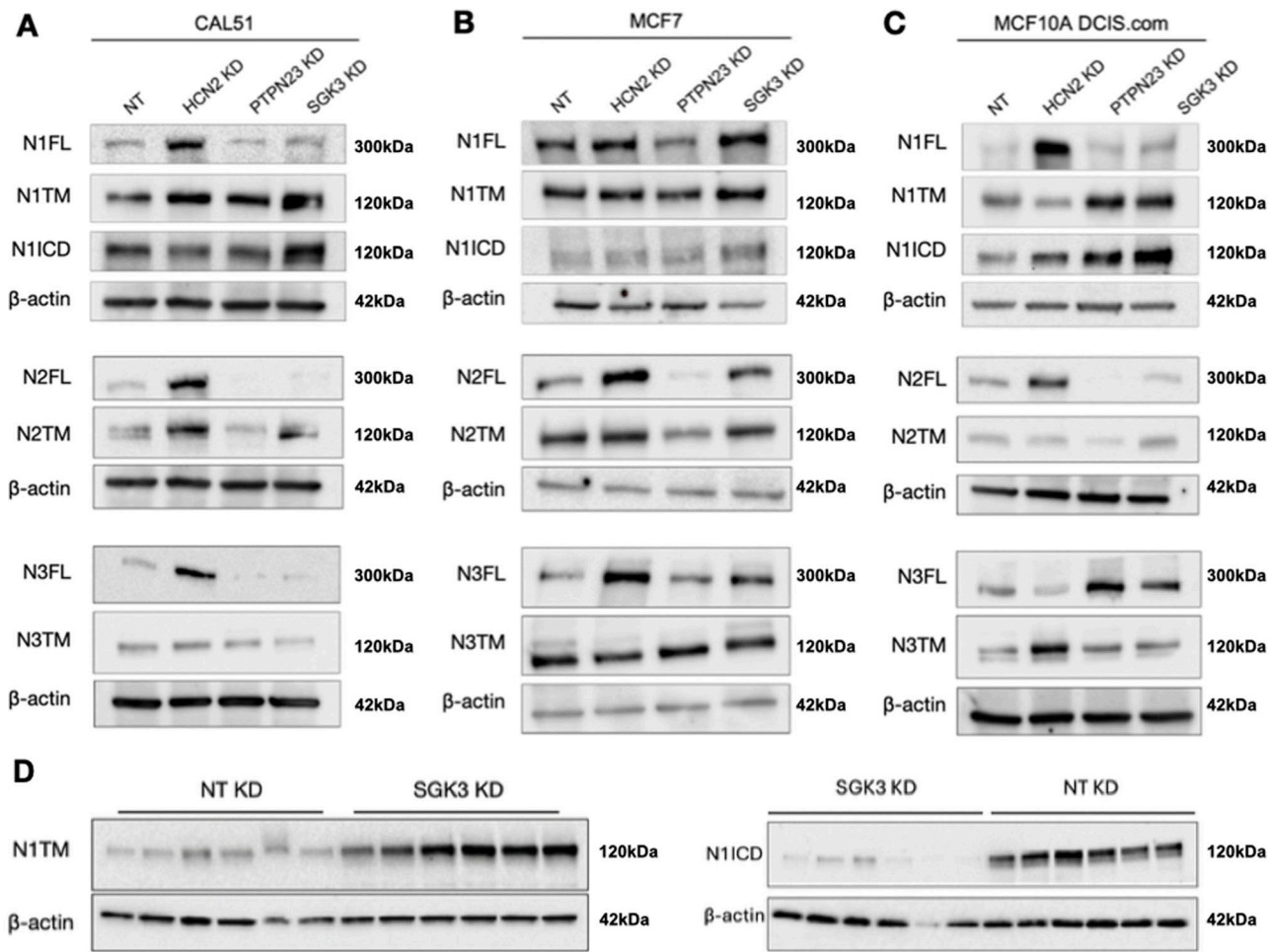

**Figure 9. NOTCH1 expression upon KD in cancer cells.**
**(A, B, C, D)** Western blot analysis of CAL-51 (A), MCF7 (B), and MCF10ADCIS.COM (C) or SCC022 (D) protein extracts treated as indicated, using antibodies against N1ICD and N1FL, N1TM. *β*-actin was used as loading control.

determination of steady-state NOTCH1 localization could uncover new genes that are important for NOTCH1 localization and associated events of activation, as well as new genes that regulate trafficking or compartment organization in MCF10A cells. We identified 51 putative genes that alter NOTCH1 localization, of which 39 resulted in modulation of signaling. We validated hits with an alternative set of reagents to limit the possibility of off target or nonspecific effects. Consistent with this, most of the identified genes are expressed in MCF10A cells according to recent RNAseq analyses (Goh et al, 2022; Gross et al, 2022). Whereas some of the candidate genes, especially those obtained upon EGTA stimulation, might be the result of indirect or combined effects with the transient extracellular $Ca^{2+}$ depletion used to activate signaling, 31, the vast majority, have been identified in unstimulated conditions, indicating that tight NOTCH1 trafficking regulation is crucial for determining basal levels of signaling. Interestingly, among these 31 basal signaling genes, 30 are required to promote RbpJk transcriptional activity, whereas only *CALM2* appeared to negatively regulate signaling. These data suggest that sustaining high Notch turnover enables basal signaling.

Depletion of *PTPN23*, *HCN2*, or *SGK3* all alter the distribution of markers of trafficking compartments, indicating that the effect on NOTCH1 trafficking and signaling of the identified candidate genes is likely to be indirect and mediated by supporting correct organization and functioning of trafficking compartments. Based on our parallel analysis of EGFR localization, we expect that a multitude of other trafficking cargoes might be affected upon depletion of PTPN23, HCN2, or SGK3. The correlative nature of our trafficking and signaling analyses does not exclude the possibility that the observed effect might be in part because of trafficking or processing alterations of Notch paralogs or ligands. In fact, we find alteration in NOTCH2 and NOTCH3 levels, similar to that of NOTCH1 upon depletion of *PTPN23*, *HCN2*, or *SGK3*. Whereas we note that depletion of NOTCH1 abates most of the signaling induced by EGTA, the relative contribution of NOTCH1 paralogs and ligands, which are known to act both in *cis* and in *trans* to regulate signaling activation, remain to be determined.

Several phenotype-based genome-wide Notch screens have been performed in vivo, in animal models such as *Drosophila* and in cellular systems. Strikingly, paralogs of 12 genes of the

51 that we have identified have been previously identified as modulators of Notch phenotypes in previous screens (*CACNB4*, *CALM2*, *CIB2*, *CMPK1*, *CTSE*, *MAP3K12*, *PIK3K2CG*, *PTPN23*, *TRPM7*, *USP36*, *UFD1L*, *ZNRF1*) (Kankel et al, 2007; Mummery-Widmer et al, 2009; Saj et al, 2010; Krämer et al, 2013; Roti et al, 2013; Ho et al, 2015). A few others have been previously involved in the regulation of Notch signaling (*ADORA1*, *CALM2*, *JAK3*, *MAP3K12*, *PLAU*, *TRPM7*) in various contexts (Wu & Sun, 2011; Na et al, 2017; Yang et al, 2020).

Four of the identified genes that have previously been associated with Notch signaling appear connected with cation regulation. We identified the vast majority in unstimulated conditions (*CALM2*, *HCN2*, *MAP3K12*, *TRPM7*), ruling out that this is because of EDTA stimulation (Mamaeva et al, 2009; Yan & Jin, 2012; Choi et al, 2013; Liu et al, 2014; Chakravarti et al, 2017; Díaz-Tocados et al, 2017; Ko et al, 2020; Wan et al, 2020; Qu et al, 2021; Tortosa et al, 2022). This evidence suggests that changes in cation regulation might fine tune extracellular and/or cytoplasmic Notch activation. Among genes that we have identified, six have previously been associated with the lysosomal mTOR pathway in the context of Notch or EGFR signaling *CIB2*, *MASTL*, *MC1R*, *SGK3*, *ZNRF1*, *ZNRF2* (Hoxhaj et al, 2016; Montero-Melendez et al, 2020; Fatima et al, 2021; Sethna et al, 2021; Shen et al, 2021; Sanz-Castillo et al, 2023), confirming that the endo-lysosomal system is an important node for Notch regulation. In addition to mTOR-associated genes, we identified as modifiers two genes that encode the deubiquitinating enzymes (DUBs) USP36 and USP39. Silencing both leads to intracellular accumulation of NOTCH1. However, *USP36* appears to promote signaling, whereas *USP39* represses it. USP39, which does not possess deubiquitinating activity and is present in the nucleus as a splicing factor (Endo et al, 2009; Wang et al, 2023), has been found to be in complex with the γ-secretase component, Nicastrin, in large-scale proteomics of human cells (Havugimana et al, 2012). USP36 is nucleolar and is involved in ribosome biogenesis (Sun et al, 2015; Fraile et al, 2018). In contrast, in flies, USP36 represses autophagy (Taillebourg et al, 2012) and has been shown to control H2B ubiquitination and chromatin regulation, leading to indirect repression of Notch signaling (Buszczak et al, 2009). This evidence suggests that these DUBs could be involved in a conserved additional layer of Notch regulation occurring in the nucleus.

Focusing on the genes most characterized in our study, we note that the relocalization of NOTCH1 to early endosomes upon *PTPN23* KD matches earlier observations in human cells that indicate that PTPN23 resides on early endosomes and that its depletion causes EGFR accumulation in EEA1-positive endosomes (Gosney et al, 2018). The activity of PTPN23 on endocytic trafficking and ESCRT function is well documented (Miura et al, 2008; Ali et al, 2013; Tabernero & Woodman, 2018). However, we find that NOTCH1 signaling activity upon *PTPN23* KD is slightly reduced, yet the loss of ESCRT components usually enhances Notch signaling (Moberg et al, 2005; Thompson et al, 2005; Vaccari & Bilder, 2005; Vaccari et al, 2008). Considering that some of the effects of *PTPN23* depletion on Notch receptors are reproduced in cancer cells, *PTPN23* has been described as a tumor suppressor as is the case of ESCRT genes, and loss of PTPN23 contributes to mammary tumorigenesis by altering endocytic trafficking of cancer-relevant cargoes (Lin et al, 2011; Manteghi et al, 2016), it will be important in the future to understand whether Notch alteration contributes to tumorigenesis initiated by loss of *PTPN23*.

We observed a major alteration of TGN morphology upon depletion of *HCN2* in MCF10A cells. This suggests that GA trafficking to the TGN might be supported by HCN2, thus leading to correct Furin cleavage of NOTCH1. Thus, inhibition of HCN2 might reduce the oncogenic effects of NOTCH1 overexpression. Interestingly, recent data indicate that HCN2 is overexpressed in triple negative breast cancer cells and that its inhibition or depletion leads to misregulation of intracellular $Ca^{2+}$ regulation and ER stress (Mok et al, 2021). Interestingly, we find elevated Notch FL levels upon *HCN2* depletion in three different types of breast cancer cell lines, including CAL-51, a triple negative cell type, suggesting that indirect regulation of Notch processing by *HCN2* might apply to many situations in breast tissue.

Finally, we have characterized the activity of *SGK3* as a negative regulator of NOTCH1 expression and signaling. *SGK3* encodes a PI3K-dependent endosome-localized serine/threonine kinase with similar substrate specificity to AKT, a kinase acting upstream of mTOR widely involved in tumorigenesis (Montero-Melendez et al, 2020). All forms of NOTCH1 accumulate in *SGK3*-depleted cells but no elevated nuclear localization was observed. We have excluded that SGK3 directly phosphorylates N1ICD or controls its stability. Thus, SGK3 is likely to act indirectly on NOTCH1 and might do so upstream of nuclear N1ICD relocalization. Because SGK3 has been proposed to promote mTOR activation in endo-lysosomal compartments, an interesting possibility is that mTOR and NOTCH1 might be alternatively regulated. Interestingly, resistance to mTOR inhibition in breast cancer cells is mediated by hVPS34 and SGK3, defining an AKT independent mTOR activation pathway (Wang et al, 2019). Considering that inhibition of Notch activity in triple negative cancer cells abrogates *SGK3* expression (Chivukula et al, 2015), a Notch/SGK3 regulatory loop might exist in breast cells that will be further investigated in the future. Consistent with this, we observe that *SGK3* depletion elevates NOTCH1 levels in breast cancer cells, including the triple negative CAL-51 line, as it does in MCF10A cells. Finally, *SGK3* depletion leads to elevated NOTCH1 levels and elevated NOTCH1 target gene expression in skin cancer cells, suggesting that SGK3 might act as a negative regulator in contexts in which NOTCH1 is a tumor suppressor, such as during skin tumorigenesis (Wan et al, 2020). In addition to *PTPN23* and *SGK3*, five other identified genes have been associated with breast tumorigenesis (*GPCR*, *GPR125*, *MET*, *NUDT5*, *TINAGL1*) (Gugger & Reubi, 1999; Stella et al, 2005; Gude et al, 2008; Xu et al, 2012; Körner et al, 2014; Zhang et al, 2014; Wright et al, 2016; Huang et al, 2018; Shen et al, 2019; Zhang et al, 2021; Sakurai et al, 2022; Spina et al, 2022; Kato et al, 2023). These studies suggest that further characterization of the identified candidate genes in breast cancer cells will prove useful.

Overall, the genes isolated in our screen expand the repertoire of factors regulating intracellular trafficking and NOTCH1 localization and signaling. They also represent potential new drug targets that may be relevant to tumorigenesis and other Notch-associated pathologies.

# Materials and Methods

## Cell culture

All cell lines were cultured at 37°C, 5% $CO_2$, in a humidified incubator. MCF10A cells (Cat #: CRL-10317; ATCC) were cultured in DMEM/F12 (Cat #: 11320033; Thermo Fisher Scientific), supplemented with 5% Horse Serum (Cat #: 16050122; Thermo Fisher Scientific), 10 mg/ml insulin (Cat #: 11376497001; Merck Life Sciences), 0.5 mg/ml Hydrocortisone (Cat #: H0888-5G; Merck Life Sciences), 100 ng/ml cholera toxin (Cat #: C8052-2MG; Merck Life Sciences), and freshly added 20 ng/ml EGF (Cat #: 90201-3; Vinci-Biochem). The MCF10A-RbpJk-Luc Notch signaling reporter cell line, which carries a luciferase gene under the control of a Notch promoter, was maintained under antibiotic selection in normal MCF10A medium supplemented with puromycin (Cat #: AG-CN2-0078-M100; Vinci-Biochem) at a final concentration of 2 mg/ml.

MCF7 cells (Cat #: HTB-22; ATCC) were cultured in DMEM (Cat #: ECB7501L; EuroClone) supplemented with 10% FBS (Cat #: A5256701; Thermo Fisher Scientific) and 1% L-glutamine (Cat #: ECB3000D; EuroClone). CAL-51 (generous gift from Dr. Daniela Tosoni, IEO, Milan) was cultured in DMEM (Cat #: ECB7501L; EuroClone) supplemented with 10% FBS (Cat #: 12103C; Merck Life Sciences) and 1% L-glutamine (Cat #: ECB3000D; EuroClone). MCF10DCIS.COM (generous gift from Dr. Daniela Tosoni, IEO, Milan) was cultured in DMEM/F12 (Cat #: 11320033; Thermo Fisher Scientific), supplemented with 5% Horse Serum (Cat #: 16050122; Thermo Fisher Scientific), 1% L-glutamine (Cat #: ECB3000D; EuroClone), 20 ng/ml EGF (Cat #: 90201-3; Vinci-Biochem), 10 $\mu$g/ml insulin (Cat #: 11376497001; Merck Life Sciences), 0.5 $\mu$g/ml hydrocortisone (Cat #: H0888-5G; Merck Life Sciences). SCC022 cells, a squamous cell carcinoma cell line, were a gift from Prof. Gian Paolo Dotto (Harvard University) and were cultured in DMEM (1X) + Glutamax (Cat #: 10566016; Thermo Fisher Scientific) supplemented with 10% FBS (Cat #: A5256701; Thermo Fisher Scientific).

## Compounds and chemicals

Where indicated, cells were treated with the following compounds and/or chemicals at indicated concentrations. $\gamma$-secretase inhibitor: DAPT (Cat #: sc-201315; Santa Cruz Biotechnology), V-ATPase inhibitor: bafilomycin A1 (BafA1) (Cat #: sc-201550; Santa Cruz Biotechnology), SGK3 inhibitor: VPS34-IN1 (Cat #: S7980; Selleckchem) and PROTAC SGK3 degrader-1 (Cat #: HY-125878; MedChem Express). Stock solutions of the compounds were prepared in cell culture grade DMSO (Cat #: EMR385250; EuroClone). $\lambda$-phosphatase ($\lambda$-PPA) along with 10X $\lambda$-PPA reaction buffer and 10X $MnCl_2$ was purchased from Santa Cruz Biotechnology (Cat #: sc-200312A). EGTA was purchased from Santa Cruz Biotechnology (Cat #: sc-3593D).

## EGTA stimulation of Notch cleavage, nuclear translocation, and signaling

Notch signaling activation was stimulated by $Ca^{2+}$ depletion, which destabilizes the non-covalent association of the extracellular portion of NOTCH1 with NOTCH1 TM (Rand et al, 2000). For short-term $Ca^{2+}$ removal, EGTA was added at a final concentration of 10 mM for up to 30 min, whereas for extended periods, EGTA was used at a final concentration of 2.5 mM for 2–4 h, followed by downstream assays.

## Immunofluorescence

Immunofluorescence (IF) was performed on cells grown on coverslips. For IF staining, cells were fixed for 10 min with 4% PFA at room temperature. They were then rinsed thrice with PBS 1X and then permeabilized for 10 min with 0.1% triton-PBS 1X or with 0.01% saponin in PBS. They were then incubated in 3% BSA in PBS 1X blocking solution or in 0.01% saponin in PBS for 30 min and then incubated with primary antibody dissolved in the appropriate blocking solution for 1 h at room temperature. Excess antibody was washed off by rinsing three times with PBS 1X, 5 min each. Samples were then incubated in secondary antibody diluted in the appropriate blocking solution for 1 h at room temperature followed by three washes using PBS 1X, 5 min per wash. The following primary antibodies were used: rat anti-full length NOTCH1, 5B5 monoclonal antibody (Cat #: SAB4200024; Merck Life Sciences) at 1:300, Goat anti-EEA1 (Cat #: sc-6415; Santa Cruz) at 1:150, rabbit anti-GIANTIN (Cat #: PRB-114C; Covance) at 1:1,000, mouse anti-EGFR 108 (hybridoma; ATCC) at 1:500, rabbit anti-reticulon-3 (RTN3; generous gift from Dr. Sara Sigismund, IEO, Milan) at 1:300, rabbit anti-TGN46 (Cat #:13573-1-AP; Proteintech) at 1:300, and rabbit anti-LAMP1 (Cat #: L1418; Merck Life Sciences) at 1:200 (requires permeabilization with saponin). Cell surface membranes were marked with phalloidin (Cat #: P5282; Merck Life Sciences) at 1:100 and the nuclei counterstained with DAPI (Cat #: D9542; Merck Life Sciences) at 1:1,000. Confocal image acquisition was performed using a Leica TCS SL confocal system. Where noted, digital images were processed using Photoshop or ImageJ software without biased manipulation.

ImageJ was used for quantitative analysis of IF images following spatial image calibration based on imaging scale bars. To assess colocalization in co-stained samples, imaging channels were first separated using the split channel function of ImageJ. To quantify NOTCH1 accumulation in the Golgi in HCN2- and SGK3-silenced conditions, and the accumulation of NOTCH1 and EGFR in the Golgi and ER upon EGTA treatment, ImageJ's freehand selection tool was used to trace regions of interest (ROIs) based on Giantin, TGN46, or RTN3 signals. The ROIs were then saved and transferred onto the NOTCH1 and EGFR channels using the ROI manager tool. Next, the areas ($\mu m^2$) and integrated densities (IntDens) of the ROIs were measured to determine organelle (GA) size and the signal intensities of NOTCH1 and EGFR (in the GA and ER), respectively. To quantify the levels of NOTCH1 and EGFR at the PM, ImageJ's freehand tool was used to mark ROIs covering the area between the inner and outer sides of the PM based on the NOTCH1 and EGFR cell surface signal. The IntDens within the ROIs was then measured to determine NOTCH1 and EGFR levels at the PM. To measure whole-cell levels of NOTCH1/EGFR, whole-cell ROIs were created by tracing the outer side of the PM based on the NOTCH1 and EGFR signals, followed by IntDens measurement to determine the levels of NOTCH1 and EGFR in whole cells. The proportions (%) of NOTCH1 and EGFR in each compartment were determined by applying the following formula in their respective channels: % IntDens = (ROI

IntDens ÷ whole-cell ROI IntDens) × 100. At least five ROIs, in at least five images were analyzed. The number of endo-lysosomal puncta, NOTCH1- and EGFR-positive puncta, and the rate of NOTCH1 colocalization with EEA1 (early endosomes) and LAMP1 (late endosomes/lysosomes) in *PTPN23-* and *SGK3*-silenced samples, as well as the rate of NOTCH1 and EGFR colocalization with LAMP1 in BafA1-treated cells were determined as follows. First, ImageJ's freehand tool was used to trace the cytoplasmic area between the edge of the nucleus and the inner side of the PM on merged (NOTCH1+DAPI+EEA1, NOTCH1+DAPI+LAMP1, or EGFR+DAPI+LAMP1) composite images. The ROIs were then saved and transferred onto the respective single channels using the ROI manager tool. The number of NOTCH1-, EGFR-, EEA1-, or LAMP1-positive puncta was then determined using ImageJ's ComDet v.0.5.5 plugin ([https://imagej.net/plugins/spots-colocalization-comdet](https://imagej.net/plugins/spots-colocalization-comdet)) after empirically determining the spot detection thresholds for each channel. For spot (puncta) detection, the intensity threshold was set at four and colocalization determined using a maximum distance of two pixels between colocalized spots. To determine the rate of colocalization with the organelle markers, the NOTCH1 or EGFR channels were merged with the EEA1 or LAMP1 channels to generate composite images and the ComDet plugin used to determine the rate of puncta colocalization. At least five ROIs in at least five images were analyzed.

## Western blot analyses

For western blot analyses, cells were scraped into 1 ml of the media they had been cultured in, transferred into prechilled 1.5 ml Eppendorf tubes, and placed on ice. They were then centrifuged at full speed (13,000*g*) for 5 min at 4°C and the pellets rinsed once with ice-cold PBS 1X. Unless indicated otherwise, cell pellets were then lysed by resuspension in 60 *µl* ice-cold RIPA buffer (Cat. #: 89901; Thermo Fisher Scientific) supplemented with a protease inhibitor cocktail (Cat #: 11697498001; Merck Life Sciences). They were then vortexed for 10 s, incubated on ice for 15 min, cleared by centrifugation at full speed for 20 min at 4°C, and the supernatants transferred into fresh 1.5 ml Eppendorf tubes on ice. Protein concentrations were quantified using a BCA protein assay kit (Cat #: 233225; Thermo Fisher Scientific) following manufacturer guidelines. Next, for each sample, 20 *µg* of the protein lysate were mixed with 7.5 *µl* of 4X Laemmli sample buffer (Cat #: 1610747; Bio-Rad Laboratories) containing 10% *β*-mercaptoethanol (Cat #: 1610710; Bio-Rad) and the volume topped up to 30 *µl* using distilled water. Depending on target protein molecular weight, the samples were then resolved on precast 4–20% Criterion TGX Stain-Free polyacrylamide gels (Cat #: 5678094; Bio-Rad) or 7.5% stain-free gels (Cat #: 4568024 or 5678024; Bio-Rad) polyacrylamide gels using 1X Tris/Glycine SDS running buffer (Cat #: 1610732; Bio-Rad). Proteins were then transferred onto 0.2 *µm* nitrocellulose membranes using Trans-Blot Turbo Mini Nitrocellulose Transfer Packs (Cat #: 1704158EDU; Bio-Rad) on a Trans-Blot Turbo TM Transfer System (Cat #: 1704150; Bio-Rad). Protein transfer was then visualized using 0.1% Ponceau S solution (wt/vol) in 5% acetic acid (Cat #: P7170-1L; Merck Life Sciences) and imaged. The membranes were then washed using 1X TBS (Cat #: 1706435; Bio-Rad) containing

0.1% Tween-20 (Cat #: 1610781; Bio-Rad) – TBST, for 15 min and then blocked with 5% skimmed milk (Cat #: 70166; Merck Millipore) in TBST for 1 h. Membranes were then incubated overnight at 4°C in the following primary antibodies: rabbit anti-cleaved NOTCH1 (Val 1744) at 1:1,000 (Cat #: 4147S; Cell signaling technology), rabbit anti-NOTCH1 (D1E11) XP at 1:1,000 (Cat #: 3608S; Cell signaling technology), rabbit anti-NOTCH2 (D76A6) at 1:1,000 (Cat #: 5732T; Cell signaling technology), rabbit anti-NOTCH3 (D11B8) at 1:1,000 (Cat #: 5276T; Cell signaling technology), mouse anti-Vinculin at 1:5,000 (Cat #: MCA4665GA; Bio-Rad), mouse anti-*β*-actin at 1:5,000 (Cat #: A-5441; Merck Life Science), or rabbit anti-Phospho-NDRG1 (Thr346) (D98G11) XP at 1:1,000 (Cat #: 5482; Cell signaling technology), and rabbit anti-SGK3 (D18D1) at 1:1,000 (Cat #:8156S; Cell signaling technology). Membranes were then washed thrice with TBST (5 min/wash) at room temperature and then incubated with HRP-conjugated goat anti-rabbit IgG (H + L) (Bio-Rad, Cat #: 170-6515) or goat anti-mouse IgG (H+L) (Cat #: 170-6516; Bio-Rad), both at 1:5,000, for 1 h at room temperature. They were then washed thrice with TBST (5 min/wash) at room temperature and signal developed for 1 min using Clarity Western ECL substrate (Cat #: 1705060; Bio-Rad), Clarity Max Western ECL Substrate (Cat #: 1705062; Bio-Rad), or Westar Hypernova ECL substrate (Cat #: XLS149,0100; Cyanagen). They were then imaged on a ChemiDoc MP imaging system (Bio-Rad). Band intensities were quantified using Image Lab software (Bio-Rad) followed by statistical analysis and graph visualization using GraphPad Prism.

## Lambda phosphatase (*λ*-PPA) treatment

For the *λ*-PPA assay, two types of RIPA lysis buffer were used – RIPA supplemented with protease (phenylmethylsulfonyl fluoride and 1 tablet of 1X cOmplete mini protease inhibitor cocktail, EDTA-free) and phosphatase (sodium fluoride and sulfur monoxide) inhibitors, and RIPA supplemented with protease inhibitors only (phenylmethylsulfonyl fluoride and 1 tablet of 1X cOmplete mini protease inhibitor cocktail, EDTA-free). The first lysis buffer (containing phosphatase inhibitors) was used as a negative control for phosphatase treatment. MCF10A cells were harvested, lysed in the appropriate ice-cold RIPA buffer (with or without phosphatase inhibitor), and protein quantified as described in the previous section. Next, for each sample (protein concentration: 1 *µg/µl*), 25 *µg* of protein (25 *µl*) were subjected to *λ*-PPA treatment following manufacturer instructions. Briefly, 0.5 *µl* of *λ*-PPA (20,000 U) were added into each tube (containing 25 *µl* of the sample), along with 3.2 *µl* of 10X *λ*-PPA buffer and 3.2 *µl* of 10X MnCl$_2$ (final volume: 32 *µl*). The samples were then mixed by gently flicking the tube and incubated for 30 min at 30°C. Next, 10.75 *µl* of 4X Laemmli loading buffer were added into each sample, followed by western blotting using the antibody against the *γ*-secretase-cleaved (Val1744) NOTCH1 intracellular domain (N1ICD).

## Cycloheximide treatment and assessment of NOTCH1 protein stability

Where indicated, after gene knockdown for 72 h, MCF10A cells were treated with 10 *µM* cycloheximide (CHX) to block protein synthesis

or an equal volume of DMSO (vehicle) for 4 h. Cells were then harvested, lysed, and protein concentration quantified as described in the HCN2 silencing traps NOTCH1 in an enlarged GA and suppresses Notch signaling section. To assess NOTCH1 protein stability, 20 µg of the protein lysate per sample were subjected to western blot analysis using the antibody against full length NOTCH1 (N1FL) and transmembrane NOTCH1 (N1TM), as well as the antibody against the γ-secretase-cleaved N1ICD. Band intensities were then analyzed on GraphPad Prism to determine the differences between the levels of N1FL, N1TM, and N1ICD in the CHX-treated versus untreated cells.

### Gene silencing

Each gene/well was targeted with a pool of four distinct siRNAs (On-Target plus SMARTpool, Dharmacon) against different sequences of the respective target transcript. Each of the 10 libraries of arrayed SMARTpools was used to knock down targets in six replicates. Three of the replicates were subjected to EGTA stimulation, whereas the other three were unstimulated. For siRNA transfection, cells were always maintained at less than six passages and trypsinized for transfection when at 60–80% confluence. All siRNA transfections were performed on black 384 well, tissue culture treated optical plates (Cat #: 3712; Corning), using the reverse transfection method. Briefly, desired siRNAs pools were diluted in Optimem (Cat #: 51985-042; Thermo Fisher Scientific) at a concentration of 50 nM, complexed with 0.12 µl of Lipofectamine RNAiMAX Transfection Reagent (Cat #: 13778500; Thermo Fisher Scientific) per 20 µl of Optimem-siRNA solution for 20 min at room temperature, and 20 µl of the complex dispensed into each 384 well. For validation of the initial 231 hits, we have used MISSION esiRNA (Sigma-Aldrich) at a final concentration of 50 nM. Cells were suspended at a density of 35 cells/µl in 2X MCF10A cell culture media and 20 µl of the cell suspension (700 cells) added into each well (containing 20 µl of the transfection complex) to obtain a final siRNA or esiRNA concentration of 25 nM. The 2X medium was prepared using each component of the medium at double its standard concentration. Double the concentration of EGF shown in the Canonical NOTCH1 signaling in human MCF10A cells section was added fresh onto the cells before they were added into the wells. Cells were then cultured for 72 h as described in the Cell culture section above, in a SteriStore automated incubator (HighRes Biosolutions). All solutions were dispensed using the Freedom EVO automated liquid handler system (Tecan).

To stimulate NOTCH1 activation, at the end of the 72 h of gene knockdown, three plates from each library were treated with EGTA for 2 h to stimulate Notch cleavage. To do this, 10 µl of fresh media containing 12.5 mM EGTA was directly added into the wells to a final concentration of 2.5 mM EGTA per well and a final volume of 50 µl/well. The NoEGTA stimulation plates received 10 µl of sterile water. The plates were then incubated for 2 h at 37°C and then fixed with 2% PFA for 15 min at room temperature. Fixation was performed by adding 50 µl of 4% PFA directly into the wells. Solutions were dispensed into the wells using a Multidrop Combi Reagent Dispenser (Thermo Fisher Scientific).

### Automated immunofluorescence analyses

Automated immunostaining for HCS was performed using a BioTek EL406 washer/dispenser equipped with a 192-tube aspiration manifold. The solution was removed from the wells and the cells rinsed once with 1X PBS. Cells were permeabilized with 0.05% triton in 1% BSA blocking solution for 1 h at room temperature followed by a single wash with 1X PBS. Cells were then incubated for 1 h at room temperature with the anti-NOTCH1 primary antibody (Cat #: SAB4200024; Merck Life Sciences) diluted at 1:350 in 1% BSA blocking solution. The primary antibody solution was then removed, and the plates washed twice with 1X PBS. Next, the cells were incubated for 1 h with 1% BSA blocking solution containing DAPI at 1:4,500 (Cat #: D9542; Merck Life Sciences), phalloidin at 1:350 (Cat #: P5282; Merck Life Sciences), and Alexa Fluor 488 anti-rat secondary antibody at 1:400 (Cat #: A21208; Thermo Fisher Scientific). They were then washed thrice with 1X PBS before imaging. Where imaging could not be performed immediately, the immunostained cells were stored at 4°C for not more than 48 h before image acquisition.

### Automated image acquisition

The 384-well plates were scanned using an automated Olympus Scan^R microscope equipped with a Hamilton arm for plate handling. Eight fields of view and three emission fluorescent channels (DAPI, phalloidin, and Alexa 488) were acquired for each well using a 20X objective. Imaging data were annotated and transferred to the Isilon infrastructure and network software and indexed by plate barcode for storage. In the annotation, each well was assigned information regarding the date of the experiment, the gene knocked down, the sub-genomic library the knocked down gene belongs to, and the treatment (EGTA versus NoEGTA). Images were then uploaded to the Columbus server (PerkinElmer), where they could be accessed for visual examination and automated analysis.

### High-content screen image analysis

An in-house Acapella (PerkinElmer) image analysis script was developed and used for batch image analysis and to quantitatively describe a set of phenotypic features. All analyzed images were first subjected to background correction and exclusion of unevenly illuminated images. Background correction was performed separately for each channel. DAPI, which was used to mask the nuclei, and phalloidin, which labeled cell surfaces, were used for cell segmentation. Segmentation was performed using a modified version of the watershed algorithm, which allowed the inversion of phalloidin channel images so as to display high pixel intensities in the cell and low intensities along the cell membrane, allowing application of the watershed approach to identify cell boundaries. This algorithm also detects and excludes regions of the image fields not covered by cells. Once the cells, their membranes, and corresponding nuclei were detected, each cell was segmented into the nucleus based on DAPI staining, the membrane based on phalloidin signal, and the cytosol (region between the nuclei and cell surface membrane).

Cells that were adjacent to field of view borders, those deemed to be too small or too big, those with saturated pixel intensities or those that were improperly segmented, were excluded from downstream image analysis. To exclude out-of-focus images, we used boxplot statistics on the distribution of intensity contrast values (on the DAPI channel) of all nuclei detected in the entire well. Using the first and the third quantiles of this distribution, we estimated the lower inferior fence (LIF) using the 95% confidence interval. For each field of view, we next established the number of nuclei presenting a contrast lower than the LIF and classified fields in which >50% of the nuclei failed to cross the LIF threshold as being out of focus. Data analysis on the EGTA-stimulated plates was performed independently of the corresponding non-EGTA–treated plates. For each gene (well), the parameters of interest were quantified and reported as z-score values (Birmingham et al, 2009).

### Candidate gene selection for validation

For each library plate, candidate genes from the EGTA and the NoEGTA conditions were considered to affect respective parameters if their knockdown shifted the z-score positively or negatively, the further the shift, the stronger the phenotype. The main parameters quantified were (a) cell viability (QC1_NoOfAnalysedCells), (b) overall Notch intensity in the cell (N4_CellNotch), (c) nuclear Notch intensity (N1_NucNotch), (d) cell surface membrane Notch intensity (N3_MembNotch), and (e) Notch accumulation in intracellular compartments (N5_PercentOfCellsWithSpots & N9_NoOfSpotsPerCell).

### Generation of the stable Notch signaling reporter cell line, MCF10A-RbpJk-Luc

A Notch pathway transcriptional output reporter cell line was generated by infecting MCF10A cells with Cignal Lenti RBPJk Reporter (luc) lentiviral particles (Cat #: CLS-014L; QIAGEN) following manufacturer instructions. This reporter system carries an inducible RBPJk-responsive firefly luciferase reporter and a puromycin resistance gene that permits the selection of cells that stably integrate the reporter construct. To isolate stably infected single cell clones, the cells were counted and serially diluted to have a final concentration of 2.5 cells/ml. Next, 200 $\mu$l of the cell suspension were seeded onto four 96-well plates so as to have 0.5 cells/well. The cell culture medium was changed twice a week and in the second week the cells were examined for single cell clones. Each of the healthiest looking clones, but with a slightly different cell morphology relative to normal MCF10A cells, was selected, transferred onto 48-well plates, grown when observing the growth characteristics, and then transferred onto six-well plates after reaching 70–80% confluence. Eventually, each single cell clone population was split into two wells of a six-well plate and frozen after reaching 70–80% confluence, followed by long term storage in liquid nitrogen. Each clone was then tested using a luciferase assay for Notch signaling inducibility using EGTA. The two best clones with the widest Notch signaling induction assay window, measured as the ratio between basal luciferase activity (in the absence of EGTA)

to activated luciferase activity (in the presence of EGTA), clone 1 and clone 13, were selected for further characterization.

To assess the ability to activate and control the Notch pathway in the stable Notch reporter cell lines, MCF10A-RbpJk-Luc single cell clones were seeded on white, opaque, flat-bottomed 96-well plates (Cat #: CLS3362-100EA; Merck Life Sciences) at 9,000 cells, in 100 $\mu$l of medium and grown overnight. Notch signaling was then stimulated by adding 11 $\mu$l of media supplemented with 25 mM EGTA into each well to reach a final concentration of 2.5 mM EGTA in 111 $\mu$l of medium. The cells were then incubated with EGTA for 2 h to allow Notch signaling activation and resulting luciferase synthesis. The luciferase assay was then performed by adding 9 $\mu$l of fresh medium containing 6.6 mM of D-luciferin into each well to have a final concentration of 500 $\mu$M D-luciferin per well, in a final volume of 120 $\mu$l per well. The cells were then incubated for 2 h after which luminescence readings were taken on a Promega GloMax multimode plate reader. To assess the ability to block Notch signaling activation in the reporter cell line, we silenced *NOTCH1*, *PSENEN*, and *ADAM10* for 72 h using siRNA (Dharmacon) before Notch signaling activation and the luciferase assay. To account for reductions in cell proliferation upon gene silencing, a cell viability assay was performed after the luciferase assay by adding 12 $\mu$l of resazurin (Cat #: TOX8-1KT; Merck Life Sciences) and the cells incubated for an additional 2 h followed by fluorescence reading on a Promega GloMax multimode plate reader.

### Secondary screen for signaling activity

Relevant genes were knocked down in MCF10A-RbpJK-Luc Notch signaling reporter cells, on white opaque 384-well plates (Cat #: 3712; Corning) followed by luciferase/proliferation assays using 500 $\mu$M D-luciferin and 2.5% resazurin after 72 h, where indicated EGTA was used at a final concentration of 2.5 mM. Briefly, a mix of D-luciferin+resazurin was prepared immediately before the experiment by dissolving D-luciferin and resazurin in normal MCF10A cell media at a concentration 3.6X higher than the desired final concentration. Where EGTA-stimulated Notch activation was desired, a mix of D-luciferin+resazurin+EGTA was prepared by including EGTA at a concentration 3.6X higher than its desired final concentration. Next, 15 $\mu$l of the appropriate solution of Notch luciferase assay mix was added into each well of the 384-well plate and the cells incubated for 4 h in normal conditions. To assess Notch signaling output (luminescence) and cell viability (resazurin fluorescence), luminescence and fluorescence were read on a PHERAstar BMG LABTECH microplate reader.

### Statistical analyses

All experimental data were analyzed on GraphPad Prism. All data represent at least three independent replicates and are shown in graphs as individual values, mean ± SD, or normalized values. Statistical differences between two groups were evaluated using unpaired two-tailed *t* tests, with *, **, ***, ****, and ns indicating $P < 0.05$, $P < 0.01$, $P < 0.001$, $P < 0.0001$, and not significant, respectively.

# Supplementary Information

# Acknowledgements

We thank Fernanda Ricci and Michela Mattioli for excellent technical help during the execution of the HCS. This work is supported by Associazione Italiana Ricerca sul Cancro (AIRC) IG grant 20661 and Worldwide Cancer Research (WCR) Grant 18-399 to T Vaccari and by IIT@SEMM.

## Author Contributions

FM Kobia: conceptualization, data curation, formal analysis, investigation, visualization, and writing—original draft, review, and editing.
L Castro e Almeida: formal analysis and investigation.
AJJ Paganoni: formal analysis and investigation.
F Carminati: data curation and investigation.
A Andronache: data curation and formal analysis.
F Lavezzari: formal analysis.
M Wade: formal analysis and supervision.
T Vaccari: conceptualization, supervision, funding acquisition, project administration, and writing—original draft, review, and editing.

## Conflict of Interest Statement

The authors declare that they have no conflict of interest.

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
