## [Reviewer comments · Life Science Alliance]

Life Science Alliance

Novel determinants of NOTCH1 trafficking and signaling in breast epithelial cells

Francis M Kobia, Luis Castro e Almeida, Alyssa JJ Paganoni, Francesca Carminati, Adrian Andronache, Francesco Lavezzari, Mark Wade and Thomas Vaccari

DOI: 10.26508/lsa.202403122

Corresponding author(s): Dr. Thomas Vaccari (University of Milan)

Review Timeline:

Submission Date:	2024-11-04
Editorial Decision:	2024-11-26
Revision Received:	2024-11-29
Accepted:	2024-12-02

Transaction Report:

Please note that the manuscript was reviewed at Review Commons and these reports were taken into account in the decision-making process at Life Science Alliance.

Reviewer #1 (Evidence, reproducibility and clarity (Required)):

In this manuscript, Kobia et al. report the results of a high-throughput screen that have been performed in a mammalian cell line to identify novel regulators of NOTCH1 trafficking and signaling. While there have been a number of screens performed in diverse model organisms and cell lines to identify novel regulators of Notch signaling, this screen is unique in that the authors used subcellular localization of endogenously expressed NOTCH1 as a primary readout, followed by assessment of their effects on signaling. They identified ~50 hits, some of which had been found in other screens and others which were novel. The authors performed additional experiments on three genes (PTPN23, HCN2 and SGK3), showing they affects different aspects of Notch trafficking and signaling. Overall, the study is well designed and the manuscript is well written.

We are pleased that the reviewer finds that “this screen is unique in that the authors used subcellular localization of endogenously expressed NOTCH1 as a primary readout, followed by assessment of their effects on signaling” and that “the study is well designed and the manuscript is well written”.

If the following comments and points are addressed, I believe the paper will be suited for publication in a good cell biology, developmental biology or cancer biology journal and have an impact on broad range of scientists and clinicians who are interested in Notch signaling and/or protein trafficking.

We are happy that the reviewer considers the manuscript, once revised, could be “suited for publication in a good cell biology, developmental biology or cancer biology journal”

Major points.

1) I see that siRNA was as a primary way to manipulate gene expression/activity. While the qRT-PCR and western blot data provided for their three hits are convincing and that significant knockdown is being achieved in the author's experiments, it is hard to judge the specificity of their manipulations. Based on the materials and methods, it seems a pool of 4 independent siRNA was used for each target gene. Were there any predicted off-targets of these siRNAs and were the experiments repeated for each hits using 4 different siRNA?

I do not see any rescue experiments or independent genetic manipulations (e.g. CRISPR knockdown or knockout) performed, so I am a bit hesitant to conclude that the effect the authors are seeing are all truly due to on-target effects of their siRNA, or there may have been some off-target (predicted or unpredicted) effects that may be interfering with their analysis.

We agree with the reviewer that an independent form of genetic validation is required to exclude potential off-target effects. In fact, we have performed the primary screen with a pool of 4 different On-Target plus siRNA (SMARTpool Dharmacon). This approach increases the likelihood of effective gene silencing and reduces sequence-specific off-targeting by diluting the concentration of each siRNA. In addition, SMARTpool design algorithm is predicted to minimize the off-target activity by using bioinformatic strategies like seed region filters and seed frequency analysis. To further reduce the possibility of off-target effects, we have performed the secondary validation leading to the identification of the final set of hits using MISSION esiRNA technology (Sigma). EsiRNAs are an independent set of reagents targeted to the set of genes emerging from the primary screen with the siRNA pools. While the On-Target plus pools are composed of 4 specific siRNA, the MISSION esiRNA are a heterogeneous mixture of siRNA that all target the same mRNA sequence. Thus, the sequences of the individual siRNA targeting the individual genes in the primary and secondary screens are different, making it extremely unlikely that the observed phenotypes are due to off target effect. Given the large number of genes involved, we have preferred to proceed with this strategy rather than opting for the possibility of deconvoluting the effects of single SMARTpool by testing separately each of the 4 siRNA in the initial On-Target plus pool. We have realized that the original description of the initial On-Target plus siRNA in the material and methods section was vague on this point and we thank the reviewer for focusing her/his attention. We now have updated the description of reagents used and the screen description (page 12 and 21).

2) Related to (1), the author used a drug called VPS34-IN as a specific inhibitor against SGK3. However, as the name suggests, VPS34-IN was developed as an inhibitor against VPS34 which is a PI3K. While Since VPS34 is known to regulate endolysosomal trafficking and likely Notch signaling, it is not clear whether the effect the authors see from this drug treatment is due to inhibition of VPS34 or SGK3 or both. The authors should explain what is known about this drug in more detail and provide an explanation on why they think the effect is purely due to inhibition of SGK3 and not due to other effects that VPS34 may have on Notch trafficking and signaling.

We concur with the reviewer that VPS34-IN might inhibit more cellular activities and not only SGK3 as indirectly demonstrated in figure 8A. To provide a direct argument, we have repeated the experiment using a proteolysis targeting chimera (PROTAC) that targets SGK3 directly. Treating cells with SGK3-PROTAC1, we now show that we obtain specific degradation of SGK3. Importantly, we also find an elevated level of N1 ICD, confirming the effect observed with VPS34-IN. These new data indicate that the elevation of N1 ICD observed upon VPS34-IN treatment is likely to be due to the inhibition of SGK3. The results replicate the findings from the genetic depletion of SGK3 and are now shown as part of fig. 8.

3) While the authors discuss the potential effect of their genetic or pharmacological manipulation on NOTCH1 trafficking, they completely ignore how their manipulations may affect the ligands. The Notch signaling activation they observe in the absence of EDTA treatment is likely due to ligand dependent signaling activation that is mediated by a ligand that is expressed in neighboring cells, especially when cultured in confluence. The authors fails to document which ligands are expressed in this cell line, and how the genetic manipulations may affect expression or trafficking of the ligands. Importantly, ligands can bind to NOTCH1 in cis as well as in trans and affects its localization and activity, and some studies suggests they do so by regulating its endolysosomal trafficking. While there is some correlation between the signaling activation and Notch trafficking shown in their data and the effect on EDTA-treated cells are likely to be cell autonomous (ligand independent), the authors should discuss the possibility of how their genetic manipulation may be affecting the ligand that is expressed in this specific cell type.

We thank the reviewer for remarking a possible role of ligands in the effects we observe. Following the reviewer suggestion, we now discuss the possibility that ligands might be affected either in cis or in trans (see updated discussion page 31).

4) The explanation of how SGK3 may affect Notch signaling is not clear. It is interesting that the amount of NICD1 that is generated doesn't necessarily correlate with the signaling output, and that SGK3 is unlikely to be affecting NICD1 phosphorylation or degradation. Can it be that this gene is regulating the translocation of NICD1 into the nucleus by somehow interacting with the nuclear import machinery or NLS of NICD1? One could perform some fractionation experiments to see whether the NICD1 that is in the nucleus and not being able to act on its targets for some reason or somehow stuck in the cytoplasm and cannot access its target.

We thank the reviewer for the suggestion that SGK3 might control N1 ICD nuclear levels. For its depletion to result in elevated signaling levels, we would expect that more N1 ICD would be found in the nucleus upon SGK3 depletion. However, as described in the text, SGK3 depletion does not significantly increase the levels of nuclear N1 ICD, neither in unstimulated conditions, nor upon EGTA treatment, which causes massive N1 ICD nuclear entry. We were able to isolate 5 out of the final 51 genes whose depletion results in elevated nuclear NOTCH1 levels with varying levels of reporter expression alterations (PHKG2, SORCS3, JAK3, UFD1L and FBXO47; supplementary file 3), however, SGK3 is not among those and its depletion consistently resulted in elevated levels of cytoplasmic and membrane associated NOTCH1 (Fig. 7A-B, supp. Fig. 7A), suggesting that SGK3 might restrict NOTCH1 cytoplasmic and membrane associated levels, rather than affecting its nuclear import. We have improved the discussion of this point (page 32-33) by taking into account the reviewer's hypothesis.

5) In Fig2, the authors show that BafA1 treatment causes NOTCH1 trafficking defects, but do not show how this affects signaling. Since this is relevant to their whole screen, I recommend the authors show how this manipulation affects not only the co-localization of NOTCH1 with EGFR and other subcellular markers but how this affects the cleavage pattern of NOTCH1 and signaling readout (luciferase and HES1/HEY1 etc mRNA levels). I believe this is an important reference point to interpret the data from the rest of the paper.

We appreciate the reviewer's insightful comment regarding the effects of BafA1 treatment on NOTCH1 signaling. In response to this suggestion and to provide a more comprehensive understanding of the impact of BafA1 on NOTCH1 signaling, we now refer more clearly in the revised text to our previous published findings (see page 19). Specifically, we tested whether BafA1 prevents Notch cleavage, nuclear entry and signaling activation. Compared to mock-treated controls, BafA1-treated cells displayed a reduction of cleavage and Notch target activation by Western blot analysis and quantitative RT-PCR (Kobia et al. Molecular Oncology, 2014). These data serve as an important reference point to interpret the findings from the rest of the paper and we thank the reviewer for suggesting to point to them in the text.

Minor points:

1) The authors say that NOTCH1, NOTCH2 and NOTCH3 are expressed in this cell line, but it seems NOTCH2 and NOTCH3 are not playing a major role here since knockdown of NOTCH1 alone seems to reduce the total NOTCH activity down to ~20% (Fig1D). The authors may want to refer to this as a justification to why they are primarily focusing on NOTCH1 here and not the other two paralogs.

We thank the reviewer for pointing out this fact that we now highlight on page 18-19.

2) I was a bit confused when I saw Fig5F since it seemed that the authors were trying to say that the reduction of NOTCH1 transcription is the major cause of Notch signaling upon PTPN23 knockdown when I just looked at this figure before reading the main text. The misinterpretation of this was due to the labeling of the Y-axis. I recommend the authors start their Y-axis here from 0 rather than 0.5, if their message was 'while NOTCH1 mRNA levels are mildly but significantly decreased upon PTPN23 knockdown, this doesn't fully explain the decrease in signaling and therefore NOTCH1 trafficking defect caused by PTPN23 is likely contributing to the signaling defect'.

We corrected as indicated.

3) It seems it may be better to move the last section of the results regarding "A secondary screen using a Notch reporter..." earlier in the paper, perhaps before discussing the three hits from this screen.

We have now moved this part to the new fig 4.

4) Page 3 line 69: Aster et al, 2017 should be cited using a number.

Corrected

5) Page 4 line 84: CSL needs to be spelled out since this is the first time this abbreviation is being used. I know it is in the glossary, but the authors should use a full name when they use an abbreviation for the first time. This goes for several other abbreviations (the authors should thoroughly go through their text).

Corrected

6) Page 4 line 89: "Su(Dx) (suppressor of Deltex)/AIP4/ITCH in mammals" should be "Su(Dx) (suppressor of Deltex, AIP4/ITCH in mammals)"

Corrected

7) The discussion portion is quite long and not organized in a logical fashion. They may want to restructure this so that they first go through what they learned from the entire screen, focus on different groups of genes they identified and their potential molecular roles in Notch signaling, and then discuss each of the hits (PTPN23, HCN2 and SGK3), following the order that they appear in the results section.

The discussion has been restructured as suggested.

Reviewer #1 (Significance (Required)):

I believe this is a unique paper with novel findings regarding regulators of Notch trafficking and signaling. The strength is that they identified potential novel regulators of this pathway. The weaknesses are that they haven't validated their findings in vivo (which could be a subject of a separate study) and the lack of some rigor regarding their shRNA and pharmacological experiments. Some further mechanistic studies could be performed, but they could be considered beyond the scope of this specific paper, which is a screen paper after all. The paper will be of interest to a rather broad audience who have interest in Notch signaling, intracellular trafficking and perhaps developmental biology as well as cancer biology. Some clinical researchers may also be interested who are studying rare genetic disorders and chemists who may be looking for potential druggable genes to target.

Reviewer #2 (Evidence, reproducibility and clarity (Required)):

SUMMARY:

Kobia and colleagues have established an MCF10a cell line-based model system in which to visualize and modulate the dynamics of endogenous NOTCH1 membrane trafficking and signaling, which they have subsequently employed to conduct an RNAi-based screen for modulators of NOTCH1 localization, stability, and signaling. They then go on to validate three of the hits from the screen: the endo-lysosomal regulator PTPN23, the channel HCN2, and the kinase SGK3.

MAJOR COMMENTS:

The design and methodology of the screen presented in this manuscript are well thought-out and adequately controlled, and the validation of the three selected hits is for the most part thorough and convincing. The authors' claims and conclusions are well-supported by the data, the statistical analysis and replicates are adequate, and the methods appear to be extremely detailed and replicable.

We are pleased that this reviewer calls our work “well thought-out”, “thorough and convincing” and “well-supported by the data”.

1) My primary concern with this report is that the entire study is based on a single tissue culture cell system, MCF10a cells. While MCF10a cells are undoubtedly a well-established and well-behaved system for studying endogenous human NOTCH1, the caveat with all tissue culture systems, particularly immortalized ones, is that they may not be representative of cells in vivo, and that the results may not be replicable either in other cell lines or in in vivo systems. Therefore, it is essential to demonstrate that the findings from this study are reproducible in additional human cell lines. Given the authors' argument that their findings may be of benefit to studies of breast cancer, I would suggest testing some of the key findings (at minimum, the VPS34-IN1 inhibitor experiment in figure 8, but ideally also some of the siRNA results from figures 5-7) in a couple of additional breast epithelial lines, whether normal or cancerous (e.g. MCF-7 or MDA-MB-231), as well as in standard cell lines derived from various other tissues (e.g. 293T or HeLa).

In response to the suggestions from this reviewer and reviewer 3 (see below), we conducted a screening of five breast cancer cell lines and selected three based on Notch receptors expression. We then performed knockdowns of the three genes in these selected cell lines and analyzed the expression of not only NOTCH1, but also NOTCH2, and NOTCH3 (following up on point 5 below) using Western blotting. We find that the most common phenotype is found upon depletion of *HCN2*, leading to the expected change of ratio between full length forms (FL) and TM forms that favors accumulation of FL. The phenotypes observed upon depletion of *SGK3* with increased FL and FL forms are also for the most part reproduced, while depletion of *PTPN23* leading to less distinctive mild decrease in FL and TM forms, are visible but less shared across the different cell lines and Notch receptors. We also show that depletion of *SGK3* leads to increased NOTCH1 levels in a skin cancer cell line in which NOTCH1 acts as tumor suppressor, indicating that the reported increase of NOTCH1 forms upon loss of *SGK3* might be not only specific to breast cells and might represent a strategy to counteract NOTCH1 loss. These additional experiments were aimed at addressing the generality of our findings across different cancer cell lines and are now presented in new fig. 9.

2) Optionally, testing the VPS34-IN1 inhibitor in primary human mammary epithelial cell lines would further strengthen the impact of this study.

We thank the reviewer for suggesting the use of primary human mammary epithelial cell lines as an alternative to breast cancer cells. While we agree that it would be important to assess phenotypes in such a system, due to lack of such expertise in the lab, we opted for analyzing a set of cancer cell lines (see point 1 above).

3) The authors state that, out of the 231 genes identified in their primary screen, only 51 were subsequently validated by the secondary screen. This seems like quite a low validation rate to me, and the authors should address the possible reasons in the text.

While 51 validated perturbations out of 231 initial seems a low rate (22%), one has to consider that to avoid off target artifact the validation round was performed using a different set of reagents (MISSION esiRNA versus On-Target plus siRNA; see also response to reviewer 1, point 1). It is our experience that with the MISSION esiRNA we achieved levels of kd that were lower than those obtained in the primary

screen. In addition, we have retained only perturbations that resulted in the same category of mislocalization, relative to the first round of screening, overall reducing the number of final candidates. We now have updated the description of reagents used and the screen description to address this point.

4) In Figure 7, given that the Notch reporter assay shows a significant change in signaling only in the presence of EGTA (Fig 7E), it seems like an oversight that the HES1, HEY1, and NOTCH1 RT-qPCR results (Figs 7F-H) are only reported for the baseline (no EGTA) condition. The authors need to report the RT-qPCR data for the +EGTA condition.

We thank the reviewer for pointing out this oversight, we have now included the data requested.

5) Although this study primarily focuses on NOTCH1, it is important to remember that NOTCH2 and NOTCH3 are also active in these cells and may contribute to some of the observed phenotypes, particularly with respect to signal output. Although I do think it would be out of the scope of this study to repeat all of the PTPN23, HCN2, and SGK3 experiments with N2 and N3, I would like to see Western blots for N2 and N3 under knockdown (or inhibitor, in the case of SGK3) conditions for these genes.

As requested, we have addressed the suggestion by performing Western blots for NOTCH2 and NOTCH3 after knocking down the three genes, both in MCF10A and in a set of cancer cell lines (see also response to point 1). The results are now included in the revised manuscript (new Fig. 9, Supp. Fig. 3-4).

6) (OPTIONAL) Given that SGK3 may have a role in regulating the endosomal component but seems to affect only levels but not localization of NOTCH1, it would be interesting to know whether EGFR behaves the same way in SGK3-siRNA-treated cells.

We thank the reviewer for the suggestion. Considering the existing data from the Alessi group indicating that SGK3 is stimulated by EGF (Malik et al 2018 PMID: 29150437), we decided to not pursue this option that might be interesting for a more detailed follow up study.

MINOR COMMENTS:

In Figure 2, it would be useful to show how the EGTA treatment and washout conditions affect Notch signaling (e.g. luciferase reporter assay and HES1/HEY1 levels).

We agree that it would be interesting to show reporter/target data upon long EGTA treatment and washout. However, the long EGTA treatment causes an initial activation of the existing NOTCH1 followed by a reduction of NOTCH1 trafficking to the plasma membrane, thus reducing the amount of NOTCH1 available at the plasma membrane for activation. Considering also that the reporter/target are expected to provide a delayed response to NOTCH1 destabilization and/or trafficking effects induced by EGTA, we think that such an experiment would provide confounding evidence and we have opted not to perform it.

In general, I found the presentation of the data clear, logical, and accurate, with a few minor points of confusion listed below. Prior studies are referenced appropriately.

In Figure 6, it would be helpful to specify what compartments the TGN46, Golgin-97, and Giantin stain respectively, and what the difference is.

We corrected the relevant text and legend as indicated. See page 24 and legend of Fig. 6.

In the Figure 7 legend, I believe that there is a typo in the line "Note that levels of N1ICD... are not comparable in C." The N1ICD levels are shown in part D, not part C.

Corrected

I found it a little confusing that the results of the Notch reporter secondary screen (Figure 9) were shown at the very end of the results section. Logically, it seems like it would fit better after Figure 4, especially given that the individual gene validations also address signaling.

We have now moved this part to the new fig 4.

Reviewer #2 (Significance (Required)):

SIGNIFICANCE:

This study identifies a number of novel modulators of NOTCH1 trafficking and stability using a tissue culture-based system. As it stands, it will primarily be of interest to the Notch signaling community, although given the importance of Notch in numerous processes such as cancer and development, other basic science researchers will likely find it useful as well. Down the line, these findings may also be of interest to clinical and translational researchers, given the potential small molecule inhibitor applications.

As a basic science Notch researcher myself with expertise and interest in high-throughput screens, I would find this study useful and complementary to my own work, and can envision many potential further directions (such as in vivo studies) that may stem from these results.

Reviewer #3 (Evidence, reproducibility and clarity (Required)):

This paper focuses on investigating the role of Notch1 receptor trafficking in Notch signal transduction in MCF10A mammalian breast epithelial cells

To identify novel genes involved in Notch trafficking, the researchers conducted a targeted siRNA screen. They used a cell-based immunofluorescence-based high-content screening to identify proteins that impact both EGTA-stimulated (artificial) and endogenous non-EGTA Notch1 activation. In a secondary screen, they validated 51 genes that affect either or both stimulation methods. The paper provides a detailed description of three of these (PTPN23, HCN2, and SGK3) that either activate or suppress Notch1 activity in MCF10A.

The experiments are well-performed and reveal interesting findings about vesicle-associated proteins and their impact on Notch localization and activity.

We are happy that the reviewer finds our experiments “well performed” and that our results “reveal interesting findings about vesicle-associated proteins and their impact on Notch localization and activity”.

However, there are several limitations to the current study, and some questions arise that could further improve the research.

1) The study is mainly descriptive, lacking new mechanistic insights into the functions of the identified proteins. The relevance of EGTA as a method for Notch stimulation is not well-explained(although widely used), especially in comparison to endogenous non-EGTA conditions.

The analysis is static and does not consider the dynamics of trafficking, despite the study's initial intention to quantitatively follow the lifetime of endogenous human NOTCH1 receptor(line 31), which would have added value.

While we provide a quantitative, time-resolved analysis of NOTCH1 exocytic and endocytic trafficking in control (Fig 2) and HCN2-depleted conditions (Supp. Fig. 6), we concur with the reviewer that the lifetime analysis has been not been performed in cells depleted of PTPN23 or SGK3. Thus, we have modified the text to limit conclusions to the observations. See revised abstract and text.

2) It's important to note that only a single cell line was used in the study, raising questions about the generality of the findings.

We have addressed this point by expanding the analyses to cancer cell lines. In response to the suggestions from this reviewer and reviewer 2 (see above), we conducted a screening of five breast cancer cell lines and selected three based on Notch receptors expression. We then performed knockdowns of the three genes in these selected cell lines and analyzed the expression of not only NOTCH1, but also NOTCH2, and NOTCH3 (following up on point 5 below) using Western blotting. We find that the most common phenotype is found upon depletion of *HCN2*, leading to the expected change of ratio between full length forms (FL) and TM forms that favors accumulation of FL. The phenotypes observed upon depletion of *SGK3* with increased FL and FL forms are also for the most part reproduced, while depletion of *PTPN23* leading to less distinctive mild decrease in FL and TM forms are visible but less shared across the different cell lines and Notch receptors. We also show that depletion of *SGK3* leads to increased NOTCH1 levels in a skin cancer cell line in which NOTCH1 acts as tumor suppressor, indicating that the reported increase of NOTCH1 forms upon loss of *SGK3* might be not only specific to breast cells and might represent a strategy to counteract NOTCH1 loss. These additional experiments were aimed at addressing the generality of our findings across different cancer cell lines and are now presented in new fig. 9.

3) Additionally, many siRNA targets vesicle components that also affect EGFR receptor localisation, which may limit the specificity of the results to Notch trafficking and demonstrate more general defects related to trafficking that also affect Notch1.

We agree with the reviewer that the effects are likely not specific for NOTCH1 but might apply to EGFR and other cargoes as well. We have modified the discussion to further clarify this point.

4) The impact of calcium depletion on EGFR and its connection to Notch signalling is not well-explained. It would be helpful to understand if EGFR has similar calcium-binding properties as Notch and if an alternative receptor unaffected by EDTA could serve as a better/additional control.

We thank the reviewer for pointing out that the EGTA dependent activation of Notch was not clearly described. We have now specified that it is commonly used to mimic ligand-dependent activation in cell culture and we provide the relevant reference in both the materials and methods and in the results. While EGFR is widely known to control Ca⁺⁺ signaling, I am not aware of it being affected by calcium chelation.

5) The relevance of the PPA-sensitive band of pNICD (Fig1) is unclear and does not align with the findings, as demonstrated by the lack of difference with SGK3 knockdown(Fig 7). This aspect is not further investigated.

We agree with the reviewer that, given the lack of difference in fig. 7J, it is not strictly necessary to show the blot in fig. 1F. However, it is useful to interpret the pattern of N1CD as shown in fig. 1E, as well as to complete the description of the behavior of endogenous NOTCH1 in MCF10A cells. It also represents a prerequisite for the analysis of fig 7J. Thus, we have decided to leave in place the analysis of fig. 1E but, following the reviewer suggestion, we have now moved the blot of fig. 7J in supp. Fig. 7.

6) The use of Hes1 mRNA as a reporter for Notch activity raises questions, particularly in Figure 5, where the PTPN23 knockdown does not affect Hes1 but affects Hey1. The authors should clarify why Hes1 is not consistently specific as a reporter and why Hey1 may be a better alternative in some experiments while using the same cells.

We have now repeated the analysis of the 3 targets (*HES1*, *HEY1*, *NOTCH1*) and, while we find a strong effect on *HEY1* expression, we find more limited but significant effects on *HES1* and *NOTCH1* expression. Please see revised Fig. 5.

7) The authors mention that most of the identified genes are expressed in MCF10A cells based on recent RNAseq analyses (line 653). Why genes recovered from the siRNA screen would not be expressed is unclear. Clarification from the authors would be beneficial.

It is possible that some of the RNA interference might not target the intended genes. While we took extensive steps to avoid such an off-target scenario for the validated set of genes (see response to reviewer 1 point 1), the fact that most of the identified genes are expressed in MCF10A might further support a functional role in regulating NOTCH1 trafficking and signaling. We have clarified this in the revised manuscript.

8) In their last figure (Fig9) the authors conduct a secondary screen and report that 39 out of 51 targets affect Notch reporter activity; from the data, it needs to be clarified if the data that is presented with the IF is consistent with the screen. In other words, hits that were affected with no EGTA are not affecting the +EGTA stimulated Notch condition and what could be an explanation for this?

Following this suggestion and that of reviewer 2 (minor comments) we have moved fig. 9 as part of the revised fig. 4. Also, we present for each gene the effect on localization and signaling in both conditions in supplemental file 3. While each perturbation would require further experimental exploration, we note that most of the depletion lead to reduced signaling output (30/39), most of which in undisturbed (no EGTA) conditions (21/30), indicating that most of the genes identified are required to support basal levels of NOTCH1 signaling. 9 depletions result in increased signaling output (9/39), the vast majority of which in conditions of stimulated signaling (+EGTA), suggesting that negative regulators might limit signaling mostly upon cleavage. We note that when we analyze the set of the initial 231 primary hits, we find that in undisturbed (no EGTA) conditions the vast majority of depletions lead to reduced signaling (110/231), with only 4/231 leading to increased signaling. In contrast, in conditions of stimulated signaling (+EGTA), approximately the same amount of depletion leads to increased or reduced signaling (45/231 increased signaling, 49/231 reduced signaling). Thus, we observe a general bias towards recovering apparent negative regulators of signaling upon EGTA stimulation. The reason for this bias is currently unknown.

Reviewer #3 (Significance (Required)):

Overall, the study provides static data without significant new mechanistic insights, showing only modest effects on Notch activity and no physiological effect (proliferation, differentiation etc..)

November 26, 2024

RE: Life Science Alliance Manuscript #LSA-2024-03122

Dr. Thomas Vaccari
Milan 20139
Italy

Dear Dr. Vaccari,

Thank you for submitting your revised manuscript entitled "Novel determinants of NOTCH1 trafficking and signaling in breast epithelial cells". We would be happy to publish your paper in Life Science Alliance pending final revisions necessary to meet our formatting guidelines.

- please be sure that the authorship listing and order is correct
- please upload your manuscript text as an editable doc file
- please upload your main and supplementary figures as single files
- please consult our manuscript preparation guidelines <https://www.life-science-alliance.org/manuscript-prep> and make sure your manuscript sections are in the correct order
- please add a separate figure legend section to your main manuscript text
- please use the [10 author names, et al.] format in your references (i.e. limit the author names to the first 10)
- please add a running title, alternate abstract, and a category for you manuscript to our system
- please add the Twitter handle of your host institute/organization as well as your own or/and one of the authors in our system
- please add a conflict of interest statement to your manuscript text
- please update the supplemental figure 4A callout; since there is only one panel for Figure S4, we don't need the designation with the panel A

Figure Check:

- please add weights next to all blots
- please add scale bars to Figure 6A and 6B

A. FINAL FILES:

B. MANUSCRIPT ORGANIZATION AND FORMATTING:

Thank you for your attention to these final processing requirements. Please revise and format the manuscript and upload materials within 5 days.

Sincerely,

Reviewer #1 (Comments to the Authors (Required)):

The authors made significant edits and performed additional experiments in response to the initial three reviewers including myself. I think the authors addressed the criticisms and suggestions pretty well. I believe the manuscript is ready to be published.

Reviewer #2 (Comments to the Authors (Required)):

The authors have satisfactorily addressed all of the questions and concerns I pointed out in my previous review. As such, I would recommend this paper be accepted for publication in LSA.

As I stated previously, the new Notch interactors identified in this study will be of interest to Notch researchers as well as those interested in trafficking and other Notch-related processes. In fact, the new data, particularly that showing the reproducibility of the findings across multiple cell types, substantially strengthens the impact.

December 2, 2024

RE: Life Science Alliance Manuscript #LSA-2024-03122R

Dr. Thomas Vaccari
University of Milan
Biosciences
via celoria 26
Milan 20133
Italy

Dear Dr. Vaccari,

Thank you for submitting your Research Article entitled "Novel determinants of NOTCH1 trafficking and signaling in breast epithelial cells". It is a pleasure to let you know that your manuscript is now accepted for publication in Life Science Alliance. Congratulations on this interesting work.

DISTRIBUTION OF MATERIALS:

Again, congratulations on a very nice paper. I hope you found the review process to be constructive and are pleased with how the manuscript was handled editorially. We look forward to future exciting submissions from your lab.

Sincerely,
